# DRIVE: Distributional and Retrieval-Augmented Bidding with Value Evaluation

**Miduo Cui** [1]  **Haochen Wang** [1]  **Shangqin Mao** [2]  **Xun Yang** [2]  **Qianlong Xie** [2]  **Xingxing Wang** [2]  **Xuri Ge** [1]
**Ying Zhou** [1]  **Zhiwei Xu** [1]

## Abstract

Auto-bidding is a core component of real-time advertising systems, where decisions must optimize long-term performance under budget and cost constraints, while online exploration is prohibitively risky. Offline reinforcement learning and, more recently, Transformer-based sequence modeling have shown promise for learning bidding policies from logged data, but their unimodal and purely parametric formulations often collapse multiple effective bidding strategies into suboptimal averaged actions and perform unreliably under sparse or long-tail traffic. To mitigate these limitations, we propose **DRIVE** (Distributional and Retrieval-Augmented Bidding with Value Evaluation), a unified Transformer-based framework that decouples candidate action generation from decision making for offline auto-bidding. DRIVE combines distributional action modeling, retrieval-augmented candidate generation from high-quality historical decisions, and value-based evaluation to select the most promising bid at inference time. Extensive experiments on AuctionNet and additional offline reinforcement learning benchmarks demonstrate that DRIVE consistently improves bidding performance and generalizes well across multiple Transformer–based methods.

## 1. Introduction

Online advertising has become a primary channel for monetizing digital traffic, where advertisers compete for impression opportunities through real-time bidding (RTB) (Yuan et al., 2013; Wang & Yuan, 2015). Modern advertising platforms widely adopt auto-bidding mechanisms (Balseiro et al., 2021a;b; Deng et al., 2021; Ou et al., 2023) to optimize long-term performance while satisfying practical con-

[1]Shandong University, Jinan, China [2]Meituan, Beijing, China. Correspondence to: Zhiwei Xu <zhiwei_xu@sdu.edu.cn>.

*Proceedings of the 43rd International Conference on Machine Learning*, Seoul, South Korea. PMLR 306, 2026. Copyright 2026 by the author(s).

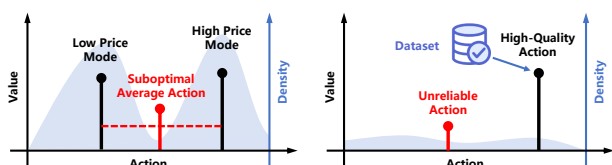

*Figure 1.* Two challenges for DT-style methods in real-world bidding. **Left:** The Average Action trap, where unimodal action modeling collapses multiple effective bidding modes into a suboptimal averaged action. **Right:** Sparse data and long-tail traffic, where current methods generate unreliable actions in low-density regions despite the presence of high-quality decisions in the dataset.

straints, such as budget limits and target Cost Per Action (CPA) (He et al., 2021; Wu et al., 2018). However, the bidding environment is inherently dynamic and uncertain, making it difficult to learn robust strategies through static heuristics or online reinforcement learning (RL) methods due to the associated risks imposed on advertisers.

Given the prohibitive risk and cost of online exploration in real-world ad auctions, offline RL (Levine et al., 2020), such as Conservative Q-Learning (CQL) (Kumar et al., 2020), becomes not just appealing but necessary. It enables the learning of bidding policies solely from logged historical data without interacting with the live market. Crucially, auto-bidding is inherently a sequential decision-making process, as current expenditures directly constrain future bidding capabilities. Consequently, Transformer-based sequence modeling approaches, such as Decision Transformer (DT) (Chen et al., 2021), have shown strong potential by leveraging long-range temporal dependencies through attention-based architectures (Vaswani et al., 2017). Building upon this line of work, a growing body of DT-style variants has recently been proposed and applied to advertising scenarios (Li et al., 2025; Gao et al., 2025). Nevertheless, directly applying such DT-style architectures to real-world bidding scenarios remains challenging, as illustrated in Figure 1. A prominent challenge is the "Average Action" trap, which arises from the fact that similar market states often admit multiple distinct yet effective bidding strategies, such as comparatively high or low bids. The unimodal or deterministic modeling in these approaches tends to collapse such diverse behaviors into suboptimal averaged actions that are neither sufficiently aggressive to secure auctions nor conservative enough to control costs. Beyond this issue, the purely parametric na-

ture of the current Transformer-based methods implies the absence of explicit mechanisms for retaining high-quality historical decisions, rendering them vulnerable to unreliable action generation under long-tail traffic or sparse data regimes.

To address these limitations, we propose **DRIVE** (**D**istributional **R**etr**I**eval-Augmented Bidding with **V**alue **E**valuation), a unified Transformer-based framework for offline auto-bidding that combines distributional action modeling, retrieval-augmented candidate generation, and value-based decision making. Unlike standard approaches, DRIVE decouples candidate action generation from decision making. Specifically, we first model the action space with a Gaussian Mixture Model (GMM) (Reynolds, 2018), enabling the policy to capture diverse yet effective bidding patterns. In addition, a retrieval mechanism encodes the current state and retrieves high-quality historical actions from similar states as supplementary candidates, providing explicit non-parametric support and mitigating unreliable actions under sparse data regimes. A value critic is further incorporated to evaluate both generated and retrieved candidates during inference and select the most promising bid. Together, these components enable DRIVE to robustly and effectively perform offline bidding. Moreover, extensive experiments across multiple settings demonstrate the effectiveness of DRIVE.

Our main contributions are summarized as follows:

- We propose **DRIVE**, a unified Transformer-based framework for auto-bidding that integrates distributional action modeling, retrieval-augmented candidate generation, and value-based evaluation.
- Extensive experiments are conducted on Auction-Net (Su et al., 2024), a representative offline bidding benchmark, demonstrating the effectiveness of DRIVE in auto-bidding scenarios.
- We further demonstrate that DRIVE is broadly applicable by seamlessly integrating it into multiple DT-style methods and consistently improving performance across a range of offline RL benchmarks.

## 2. Related Work

### 2.1. Evolution of Auto-bidding Strategies

Early research in auto-bidding primarily relied on static optimization or control-theoretic frameworks. Heuristic bidding strategies, ranging from linear (Perlich et al., 2012) to non-linear functions (Zhang et al., 2014), derive bid prices based only on the predicted value of each impression, such as predicted click-through rate (pCTR). To account for budget constraints, control-based methods, including PID controllers (Chen et al., 2011; Lee et al., 2013; Yang et al., 2019) and Smart Pacing (Xu et al., 2015), were developed

to smooth consumption. However, these approaches are inherently myopic: they focus on immediate returns or pre-defined heuristic rules, failing to optimize for long-term objectives in the highly stochastic auction environment.

To overcome the myopia of static strategies, reinforcement learning (RL) was introduced to model bidding as a sequential decision process. Cai et al. (2017) pioneered a model-based framework by casting bidding as a constrained Markov decision process (MDP) (Puterman, 1990). However, approaches that rely on explicit environment modeling often incur substantial computational overhead and suffer from simulation-to-reality discrepancies (Wu et al., 2018). As a result, subsequent research shifted toward model-free RL paradigms (Wu et al., 2018). Notably, Liu et al. (2020) proposed a dynamic strategy leveraging the TD3 algorithm (Fujimoto et al., 2018) to optimize continuous bidding factors directly, bypassing the need for complex market modeling. In real-world bidding systems, however, online RL is generally impractical, as exploratory actions may incur substantial financial costs. Consequently, offline RL (Levine et al., 2020), which learns policies solely from logged historical data, has emerged as a practical and dominant paradigm for auto-bidding.

### 2.2. Offline Reinforcement Learning for Auto-bidding

Despite its practical appeal, offline RL introduces fundamental challenges in auto-bidding scenarios. A central issue is distribution shift, where learned policies may exploit actions that are poorly supported by the logged data, leading to unreliable value estimation and unsafe decisions. To address this issue, prior work has proposed conservative or in-sample learning methods, including BCQ (Fujimoto et al., 2019), CQL (Kumar et al., 2020), and IQL (Kostrikov et al., 2022), which aim to mitigate overestimation on out-of-distribution (OOD) actions (Levine et al., 2020).

While the above value-based offline RL methods are effective at addressing OOD overestimation, they often struggle with long-horizon credit assignment and complex sequential dependencies. This limitation has motivated a paradigm shift toward reformulating RL as generative sequence modeling (Janner et al., 2021; Chen et al., 2021). Notably, Decision Transformer (DT) (Chen et al., 2021) leverages the self-attention mechanism to generate actions conditioned on desired future returns, effectively capturing long-range dependencies. Building on this framework, recent methods have sought to integrate value information into generative policies. For example, GAVE (Gao et al., 2025) introduces value-guided exploration during training, while GAS (Li et al., 2025) employs post-training search with multi-critic voting to refine actions. Peak-Return Greedy Slicing (Xu et al., 2026) offers a data-centric paradigm for improving Transformer-based offline RL, where high-return subtrajec-

tories are selected to construct more informative training sequences. Beyond Transformer-based models, DiffBid (Guo et al., 2024) employs conditional diffusion to model bidding distributions. However, aside from the prohibitive inference latency caused by iterative sampling, it struggles to effectively learn the reverse diffusion process in highly dynamic and long-horizon environments, leading to inaccurate trajectory prediction and suboptimal policy performance.

Despite these advances, DT-style generative approaches remain limited in real-world bidding scenarios. They typically rely on unimodal regression objectives and point-estimate decoding, which fail to capture the inherently multimodal nature of optimal bidding behaviors. As a result, multiple distinct yet effective bidding strategies are often collapsed into averaged actions, leading to suboptimal performance. In contrast, DRIVE explicitly models the action distribution to preserve diverse bidding modes, enabling more robust and effective decision-making under complex and uncertain market conditions.

### 2.3. Retrieval-Augmented Decision Making

Retrieval-Augmented Generation (RAG) was introduced in natural language processing (NLP) to mitigate hallucinations and outdated knowledge in parametric models by incorporating evidence retrieved from large external corpora (Lewis et al., 2020; Guu et al., 2020; Borgeaud et al., 2022). By grounding generation in retrieved documents, RAG improves both factual accuracy and interpretability in knowledge-intensive tasks like open-domain question answering. Motivated by these benefits, retrieval mechanisms have recently been adopted in RL to better exploit past experience. DT-Mem (Kang et al., 2024) augments Decision Transformers with an internal memory to reduce forgetting in multi-task settings, while RA-DT (Schmied et al., 2024) retrieves relevant subtrajectories from an external index to extend context length for long-horizon decision making. These studies suggest that retrieval can serve as an explicit non-parametric component, enhancing decision quality by reusing high-quality historical experiences. Inspired by this line of work, our method leverages retrieval to enhance the robustness of bidding decisions under sparse and long-tail data regimes.

## 3. Preliminaries

### 3.1. RTB Environment and Optimal Bidding

Consider an advertising campaign consisting of $N$ sequential impression opportunities in a real-time bidding (RTB) environment with a generalized second-price (GSP) auction mechanism (Lucier et al., 2012). For each impression $i$, the advertiser submits a bid $b_i$. The winning outcome of the auction is represented by a binary indicator $x_i \in \{0, 1\}$, and

the corresponding payment is denoted by $c_i$, which equals the second-highest bid. Each impression is associated with a value $v_i$, such as a click or conversion. The advertiser aims to maximize the total accumulated value $\sum_{i=1}^{N} v_i x_i$ subject to a total budget constraint $B$ and a set of key performance indicator (KPI) constraints, such as cost-per-action (CPA) or return on investment (ROI). This objective can be formulated as the following constrained optimization problem:

$$\max_{\{x_i\}_{i=1}^{N}} \quad \sum_{i=1}^{N} v_i x_i$$
$$\text{s.t.} \quad \sum_{i=1}^{N} c_i x_i \leq B, \tag{1}$$
$$\mathcal{G}_j(x_{1:N}) \leq \mathcal{K}_j, \quad \forall j.$$

where $\mathcal{G}_j(\cdot)$ denotes the constraint function corresponding to the $j$-th KPI, and $\mathcal{K}_j$ specifies its target threshold. Previous studies (He et al., 2021; Zhang et al., 2014) have shown that, under mild assumptions, the optimal bidding strategy can be derived from the Karush–Kuhn–Tucker (KKT) conditions and admits a unified affine form. In practice, this strategy is often simplified to a scaled value-based bidding rule:

$$b_i^* = \lambda \, v_i, \tag{2}$$

where $\lambda$ is a control parameter determined by the Lagrange multiplier associated with the budget and KPI constraints. As a result, modern auto-bidding methods commonly focus on dynamically adjusting $\lambda$ to adapt to stochastic market conditions and evolving constraints.

### 3.2. Sequential Decision-Making for Auto-bidding

In this paper, the auto-bidding problem is formulated as a sequential decision-making task modeled by a Markov Decision Process (MDP) $\mathcal{M} = \langle \mathcal{S}, \mathcal{A}, \mathcal{P}, \mathcal{R}, \gamma \rangle$. An episode corresponds to a bidding cycle, typically one day, which is discretized into $T$ time steps. At each time step $t$, the agent determines a bidding control decision $a_t \in \mathcal{A}$ that applies to all impressions arriving within the corresponding interval. The policy depends on the current state $s_t \in \mathcal{S}$, expressed as $\pi(a_t \mid s_t)$. State transitions follow the environment dynamics $\mathcal{P} : \mathcal{S} \times \mathcal{A} \to \mathcal{S}$, and upon transitioning to the next state $s_{t+1}$, the environment provides a scalar reward $r_t$ reflecting the performance contribution achieved during time step $t$. $\mathcal{R} : \mathcal{S} \times \mathcal{A} \to \mathbb{R}$ is the reward function, and $\gamma \in (0, 1]$ is the discount factor.

**State.** The state $s_t$ summarizes the contextual information available at time step $t$ from both campaign-level and market-level perspectives. Campaign-level features characterize the internal status and global constraints of the advertiser. Market-level features capture the external auction environment and its temporal dynamics, which are derived from aggregated historical auction observations.

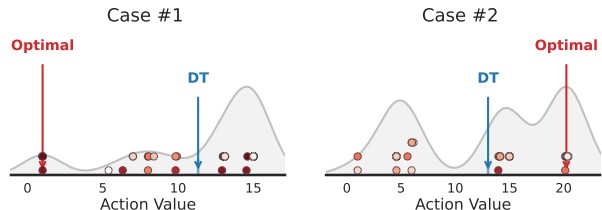

*Figure 2.* Real failure cases in AuctionNet. The blue line denotes the suboptimal predicted average action, the red line the optimal action, and colored points indicate dataset actions with color intensity reflecting RTG.

**Action.** The action $a_t$ specifies a bid adjustment factor, denoted by the bidding parameter $\lambda_t$ in Equation (2). This parameter scales the predicted value $v$ of each impression, which is assumed to be available from a pre-trained prediction model, to compute the final bid price.

**Reward.** The reward $r_t$ measures the contribution of the agent's decision $a_t$ at time step $t$ to the advertiser's objective. Typical reward definitions include the total conversion value or the number of clicks obtained during the corresponding interval, optionally combined with penalty terms to reflect budget or KPI violations.

To leverage the sequence modeling capabilities of Transformer-based offline RL methods, such as DT, the offline dataset is organized into trajectories of the form $\tau = (\hat{R}_0, s_0, a_0, \ldots, \hat{R}_T, s_T, a_T)$. $\hat{R}_t = \sum_{i=t}^{T} \gamma^{i-t} r_i$ denotes the return-to-go (RTG), which conditions action generation on desired future performance. During training, the Transformer learns to predict actions conditioned on the RTG-state context from offline trajectories. At inference time, the policy generates actions autoregressively based on the current state and target RTG, enabling long-horizon credit assignment through sequence modeling.

## 4. Methodology

In this section, we present **DRIVE**, a unified Transformer-based framework for auto-bidding. DRIVE extends standard Transformer-based offline RL by incorporating three key components: **(I)** a distributional action head for capturing the multimodal bidding behaviors, **(II)** a retrieval-augmented mechanism that grounds decisions in relevant high-quality historical trajectories, and **(III)** a value-based critic to decide the final action for improving robustness. For clarity and completeness, we provide a comprehensive table of notations in Appendix A.1 and the detailed algorithmic workflow in Appendix A.2.

### 4.1. GMM-Based Action Generation

Transformer-based offline RL models generate actions via conditional sequence modeling:

$$a_t \sim \pi(a_t \mid \tau_{0:t-1}, \hat{R}_t, s_t). \tag{3}$$

Most existing Transformer-based approaches in continuous action spaces adopt a deterministic regression head optimized with a mean squared error (MSE) objective (Wang & Bovik, 2009). Such unimodal regression tends to average over diverse historical actions, as illustrated in Figure 2 with real-world examples from AuctionNet. This issue is particularly pronounced in bidding environments, where conservative and aggressive strategies coexist, often resulting in collapsed and non-informative actions.

To explicitly capture multimodal bidding behaviors, we replace the deterministic action head with a Gaussian Mixture Model (GMM) head (Reynolds, 2018). GMMs, following the Mixture Density Network paradigm (Bishop, 1994), model conditional action distributions by predicting a set of $M$ mixture components:

$$\{\alpha_m, \mu_m, \sigma_m^2\}_{m=1}^{M}. \tag{4}$$

$\alpha_m \in [0, 1]$ denotes the mixing coefficient of the $m$-th component, satisfying $\sum_{m=1}^{M} \alpha_m = 1$, while $\mu_m$ and $\sigma_m^2$ represent the mean and variance of the corresponding Gaussian component, respectively. All mixture parameters are dynamically predicted conditioned on the current trajectory context. The resulting action distribution is then given by:

$$P(a_t \mid \tau_{0:t-1}, \hat{R}_t, s_t) = \sum_{m=1}^{M} \alpha_m \mathcal{N}(a_t \mid \mu_m, \sigma_m^2), \tag{5}$$

which forms a multi-peaked density capable of representing distinct bidding modes. This GMM-based variant is trained by maximizing the log-likelihood of historical actions from the offline dataset $\mathcal{D}$:

$$\mathcal{L}_{\text{GMM}} = -\mathbb{E}_{\tau \sim \mathcal{D}} \left[ \sum_{t=1}^{T} \log \left( \sum_{m=1}^{M} \alpha_m \mathcal{N}(a_t \mid \mu_m, \sigma_m^2) \right) \right]. \tag{6}$$

This distributional objective enables the policy to represent multiple bidding strategies simultaneously, rather than collapsing them into a single point estimate.

**Inference-Time Sampling.** Unlike deterministic policies that output a single action, the GMM-based action head enables stochastic sampling at inference time. Given the predicted mixture parameters, a set of candidate actions $\mathcal{A}_{\text{gen}} = \{a_t^{(l)}\}_{l=1}^{L}$ is generated by sampling from the learned mixture distribution:

$$a_t^{(l)} \sim \sum_{m=1}^{M} \alpha_m \mathcal{N}(\mu_m, \sigma_m^2). \tag{7}$$

This sampling mechanism preserves multiple plausible bidding modes and yields a diverse candidate pool for subsequent evaluation.

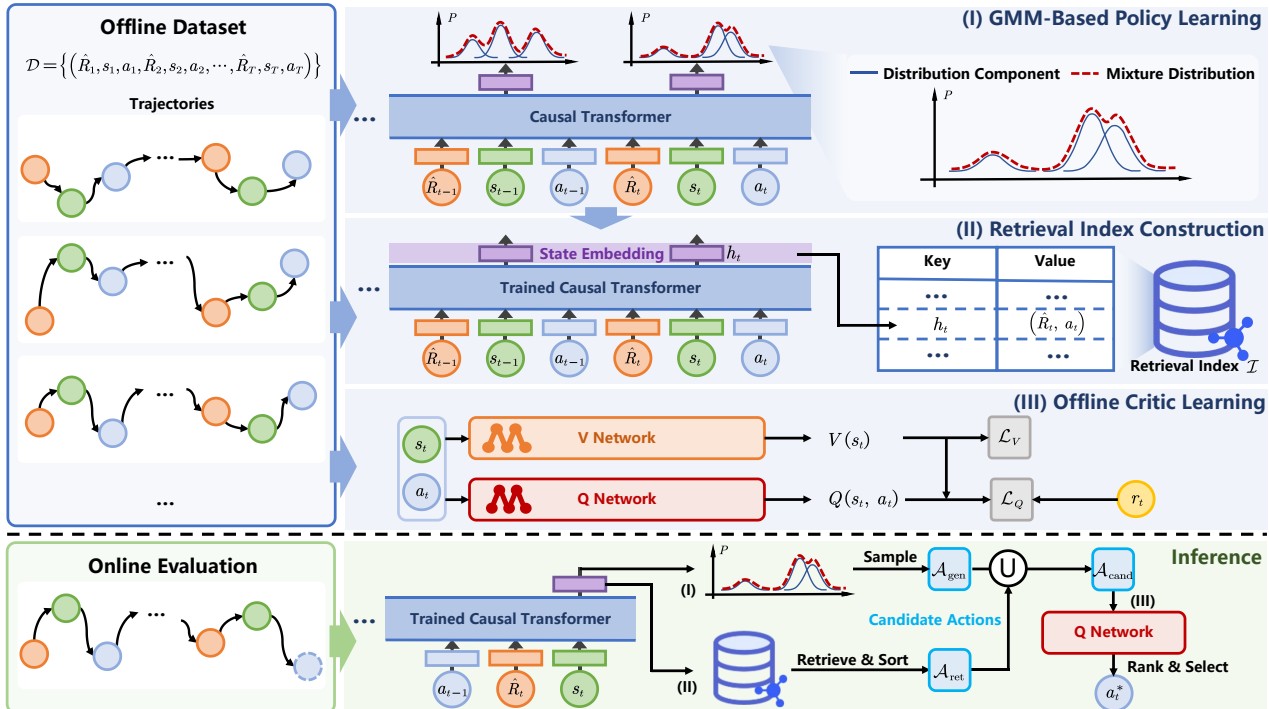

*Figure 3.* The DRIVE framework. DRIVE is built upon a Transformer-based offline RL paradigm, incorporating (I) a multimodal GMM-based policy, (II) a retrieval index over contextual state embeddings, and (III) a value-based offline critic. At inference time, generated and retrieved actions are jointly evaluated, and the final action is selected via critic-based ranking.

## 4.2. Retrieval-Augmented Candidate Generation

Retrieval-augmented generation (Lewis et al., 2020) improves robustness under sparse inputs by grounding parametric models with high-quality samples from the training data. Motivated by this property, we incorporate a retrieval mechanism into the decision-making process to augment the parametric Transformer-based policy with relevant examples drawn from the offline dataset. This design enables the policy to anchor its decisions to previously observed high-performing behaviors, particularly in sparse and long-tail bidding scenarios. Specifically, the GMM-based Transformer encoder is first used to encode the offline dataset into contextual state embeddings, and at inference time, high-quality actions are retrieved based on state similarity to the current decision context.

**Retrieval Index Construction.** We construct a retrieval index $\mathcal{I}$ from the offline training dataset using contextual state embeddings. While the pre-trained policy encoder can be reused to minimize overhead, for large-scale industrial tasks, we employ a dedicated lightweight Transformer encoder. This design reduces embedding dimensions for efficient search without compromising policy capacity. Rather than storing raw states, each trajectory in the offline dataset $\mathcal{D}$ is processed by the encoder to obtain a contextual embedding at every time step:

$$h_t = f_{\text{enc}}(\tau_{0:t-1}, \hat{R}_t, s_t) \in \mathbb{R}^d, \qquad (8)$$

where $f_{\text{enc}}(\cdot)$ denotes the contextual encoder, and $h_t$ corresponds to the hidden representation of the state before the action head. This embedding captures both temporal dependencies and semantic context.

To improve retrieval efficiency and candidate quality, optional lightweight filtering can be applied during index construction to remove low-quality or uninformative transitions, depending on practical requirements. For each retained transition, the contextual embedding $h_t$ is used as the retrieval key, while the corresponding action $a_t$ and Return-to-go $\hat{R}_t$ are stored as aligned values. Similarity search is performed directly in this embedding space. Additional implementation details of the retrieval module are in Appendix A.3.

**Inference-Time Retrieval.** At inference time, the current decision context at time step $t$ is encoded by the Transformer encoder to obtain the contextual state embedding $h_t$, as defined in Equation (8). To retrieve actions that are both contextually relevant and high-performing, we adopt a retrieve-then-filter strategy. First, a candidate pool of $K_{\text{pool}}$ nearest neighbors is retrieved from the index $\mathcal{I}$ based on cosine similarity (Xia et al., 2015):

$$\mathcal{C}_{\text{pool}} = \left\{ (a_k, \hat{R}_k) \mid k \in \text{Top-}K_{\text{pool}}^{\text{sim}}(\mathcal{I}, h_t) \right\}, \qquad (9)$$

where $\text{Top-}K_{\text{pool}}^{\text{sim}}$ returns the indices of the $K_{\text{pool}}$ entries in $\mathcal{I}$ with the highest cosine similarity to $h_t$. The retrieved candidates are then ranked according to their stored RTG

values, and the top-$K$ actions are selected:

$$\mathcal{A}_{\text{ret}} = \{ a_k \mid k \in \text{Top-}K^{\hat{R}}(\mathcal{C}_{\text{pool}}) \}. \qquad (10)$$

Top-$K^{\hat{R}}$ returns the indices of the $K$ candidates with the highest stored RTG values. These retrieved actions complement the generated candidates by providing high-quality references from the training data, improving decision robustness under sparse and long-tail bidding conditions.

### 4.3. Value-Based Action Evaluation

While the GMM head provides diverse probabilistic candidates and the retrieval module supplies high-quality references from the dataset, relying on either source alone can be risky. Generative candidates capture multimodal bidding behaviors but may suffer from model uncertainty, whereas retrieved actions offer stability but can be suboptimal when the current context differs from past observations. To robustly select the final bid, we introduce a value-based critic to evaluate all candidate actions before execution.

A broad class of value-based offline RL methods can be applied to train the critic for evaluating candidate actions and determining the final bid. In this work, the critic follows the Implicit Q-Learning (IQL) paradigm (Kostrikov et al., 2022), which estimates action values strictly within the support of the offline dataset without explicitly penalizing unseen actions. Specifically, two Q-functions and a state value function are learned from offline data. The state value function is trained to approximate an upper expectile of the Q-value distribution, enabling implicit maximization over in-sample actions. This is achieved via expectile regression (Jiang et al., 2017):

$$\mathcal{L}_V = \mathbb{E}_{(s,a)\sim\mathcal{D}} \Big[ L_2^{\eta} \big( \min_{i=1,2} Q_i(s,a) - V(s) \big) \Big], \qquad (11)$$

where $L_2^{\eta}(u) = |\eta - \mathbb{I}(u < 0)| u^2$ and $\eta \in (0.5, 1)$ controls the degree of implicit maximization. The Q-functions are then trained using a Bellman target constructed from the learned state value:

$$\mathcal{L}_Q = \mathbb{E}_{(s,a,r,s')\sim\mathcal{D}} \Big[ \big( Q(s,a) - (r + \gamma V(s')) \big)^2 \Big], \quad (12)$$

which ensures strictly in-sample learning and provides a stable critic for offline evaluation.

Crucially, the critic adapts to task requirements by shaping the reward to reflect safety constraints. In unconstrained settings, the raw reward $r$ is used, whereas for constrained tasks it is replaced in Equation (12) with a constraint-aware reward $r'$. Specifically, for CPA-constrained tasks, the shaped reward is defined as:

$$r' = r \times \min \Big( 1, \Big( \frac{\mathcal{K}}{C + \epsilon} \Big)^{\beta} \Big), \qquad (13)$$

where $\mathcal{K}$ denotes the target CPA threshold, $C$ represents the realized CPA, and $\beta = 2$ controls the penalty steepness. This shaping ensures the learned value landscape inherently reflects safety, guiding the agent toward feasible regions.

**Inference-Time Decision-Making.** At inference time, DRIVE generates the final decision by invoking the above three modules. Specifically, it first samples $L$ candidate actions $\mathcal{A}_{\text{gen}}$ from the GMM-based policy to cover diverse bidding modes. In parallel, a set of $K$ high-quality actions $\mathcal{A}_{\text{ret}}$ is retrieved using the RTG-guided retrieval strategy described earlier. The two sets are then combined into a unified candidate pool $\mathcal{A}_{\text{cand}} = \mathcal{A}_{\text{gen}} \cup \mathcal{A}_{\text{ret}}$. The final action is selected by evaluating all candidates with the learned critic:

$$a^* = \arg \max_{a \in \mathcal{A}_{\text{cand}}} \min_{i=1,2} Q_i(s, a). \qquad (14)$$

This inference procedure integrates the diversity of generative candidates, the reliability of retrieved actions, and value-based evaluation to produce robust decisions. Through this design, DRIVE can be integrated into other Transformer-based offline RL algorithms, providing a general solution to the average-action issue and unreliable decisions under long-tail and sparse data regimes, particularly in auto-bidding.

## 5. Experiment

Extensive experiments are conducted to evaluate the proposed DRIVE framework. The primary objective is to examine its ability to alleviate the "Average Action" issue in multimodal decision landscapes and to improve decision reliability under sparse and long-tail data regimes in auto-bidding scenarios. In addition, representative offline RL tasks are used to assess its general effectiveness and transferability beyond bidding. A comprehensive ablation study further analyzes the contribution of each core component. Finally, DRIVE is integrated into multiple Transformer-based offline RL architectures to demonstrate its plug-and-play generality. Detailed experimental settings and hyperparameters are provided in Appendix A.4 for reproducibility.

**Datasets.** The proposed framework is evaluated on *Auction-Net* (Su et al., 2024) and the *D4RL* benchmark (Fu et al., 2020). AuctionNet is a large-scale industrial benchmark constructed from real-world bidding logs, featuring highly stochastic dynamics and strict budget and CPA constraints. Both dense and sparse variants are used to assess robustness under challenging bidding conditions. To evaluate generalization, D4RL benchmarks including Gym-MuJoCo and Maze2D are adopted, covering continuous control and long-horizon navigation under varying data quality. Additional dataset statistics and feature descriptions are provided in Appendix B.1.

**Baselines.** The baselines are grouped into three categories to cover representative algorithmic paradigms in offline RL

*Table 1.* Comparison with baselines on AuctionNet and AuctionNet-Sparse datasets under different budget constraints. We report values (mean ± standard deviation) over 10 seeds. The best results are **bolded**, and the second-best results are underlined.

| Dataset | Budget | CQL | IQL | BCQ | DiffBid | DT | CDT | GAS | GAVE | GAVE-S | DRIVE |
|---|---|---|---|---|---|---|---|---|---|---|---|
| AuctionNet | 50% | 212 ± 3.06 | 194 ± 2.07 | 181 ± 3.26 | 155 ± 2.59 | 208 ± 1.75 | 208 ± 2.06 | 200 ± 2.68 | 133 ± 1.69 | 108 ± 1.36 | **212 ± 1.57** |
| | 75% | **300 ± 2.65** | 284 ± 1.76 | 263 ± 3.39 | 225 ± 1.47 | 298 ± 2.21 | 300 ± 2.00 | 295 ± 3.33 | 192 ± 1.74 | 158 ± 1.49 | 297 ± 2.25 |
| | 100% | 382 ± 2.25 | 366 ± 1.82 | 343 ± 8.2 | 285 ± 1.85 | 373 ± 3.18 | 382 ± 3.19 | 381 ± 2.71 | 245 ± 1.00 | 209 ± 2.39 | **399 ± 3.74** |
| | 125% | 463 ± 2.61 | 444 ± 2.30 | 414 ± 12.34 | 334 ± 2.80 | 430 ± 2.98 | 450 ± 2.68 | 457 ± 3.07 | 298 ± 2.05 | 261 ± 2.52 | **475 ± 5.33** |
| | 150% | 535 ± 2.97 | 500 ± 2.57 | 478 ± 8.65 | 377 ± 3.05 | 477 ± 2.12 | 508 ± 2.67 | 525 ± 3.24 | 350 ± 2.10 | 316 ± 2.61 | **551 ± 4.64** |
| **Avergae** | | 378.4 | 357.6 | 335.8 | 275.2 | 357.2 | 369.6 | 371.6 | 243.6 | 210.4 | **386.6** |
| AuctionNet Sparse | 50% | 20.2 ± 0.69 | 17.9 ± 0.63 | 17.9 ± 0.38 | 14.9 ± 0.60 | 15.8 ± 0.57 | 17.8 ± 0.65 | 14.2 ± 0.68 | 8.7 ± 0.36 | 17.6 ± 0.66 | **20.4 ± 0.44** |
| | 75% | **28.8 ± 0.72** | 26.9 ± 0.66 | 26.7 ± 0.62 | 20.2 ± 0.68 | 23.1 ± 0.22 | 26.9 ± 0.53 | 20.8 ± 0.84 | 9.8 ± 0.41 | 25.7 ± 1.09 | 27.8 ± 0.53 |
| | 100% | 37.1 ± 0.59 | 35.2 ± 1.09 | 34.2 ± 0.76 | 24.3 ± 0.54 | 30.6 ± 0.69 | 35.9 ± 0.68 | 27.1 ± 0.83 | 9.9 ± 0.43 | 34.3 ± 1.04 | **37.3 ± 0.87** |
| | 125% | **44.6 ± 1.12** | 43.7 ± 0.77 | 42.0 ± 1.05 | 28.2 ± 0.75 | 37.9 ± 0.54 | 44.1 ± 1.09 | 33.1 ± 0.67 | 9.9 ± 0.45 | 41.3 ± 1.14 | 43.1 ± 1.18 |
| | 150% | 49.6 ± 1.27 | 51.4 ± 0.97 | 47.7 ± 0.88 | 31.4 ± 0.61 | 45.7 ± 0.89 | 50.6 ± 1.61 | 40.2 ± 0.83 | 10.0 ± 0.46 | 49.2 ± 0.65 | **51.8 ± 0.53** |
| **Avergae** | | 36.06 | 35.02 | 33.7 | 23.8 | 30.62 | 33.06 | 27.08 | 9.66 | 33.62 | **36.08** |

*Table 2.* Comparison with baselines on D4RL benchmarks. We report normalized scores over 5 seeds. The best results are **bolded**, and the second-best results are underlined. The results for baselines are taken from original papers.

| Domain | Dataset | CQL | IQL | BEAR | TD3+BC | BC | DT | PDiT | DRIVE |
|---|---|---|---|---|---|---|---|---|---|
| Gym MuJoCo | halfcheetah-expert | 62.4 | 86.7 | 53.4 | 90.7 | 86.2 ± 9.4 | 91.7 ± 0.3 | 73.0 ± 4.3 | **93.0 ± 0.9** |
| | hopper-expert | 111.0 | 91.5 | 96.3 | 98.0 | 67.5 ± 13.1 | 109.8 ± 0.5 | **111.4 ± 0.1** | 106.1 ± 1.2 |
| | walker2d-expert | 98.7 | 109.6 | 40.1 | **110.1** | 108.7 ± 0.3 | 108.9 ± 0.1 | 108.8 ± 0.4 | 108.6 ± 0.1 |
| | halfcheetah-medium | 44.4 | 47.4 | 41.7 | **48.4** | 40.5 ± 0.1 | 40.0 ± 0.1 | 42.8 ± 2.3 | 46.7 ± 0.1 |
| | hopper-medium | 58.0 | 66.3 | 52.1 | 59.3 | 59.9 ± 1.5 | 63.6 ± 2.6 | **68.2 ± 2.4** | 67.3 ± 1.5 |
| | walker2d-medium | 79.2 | 78.3 | 59.1 | 83.7 | 78.8 ± 1.1 | 78.1 ± 1.5 | 77.6 ± 0.6 | 81.0 ± 0.1 |
| | halfcheetah-medium-replay | **46.2** | 44.2 | 38.6 | 44.6 | 35.8 ± 0.7 | 35.0 ± 1.0 | 40.8 ± 2.3 | 43.7 ± 0.2 |
| | hopper-medium-replay | 48.6 | 94.7 | 33.7 | 60.9 | 48.0 ± 28.2 | 78.7 ± 0.3 | **89.6 ± 2.7** | 89.2 ± 2.1 |
| | walker2d-medium-replay | 26.7 | 73.9 | 19.2 | 81.8 | 57.5 ± 3.3 | 71.5 ± 1.7 | 74.1 ± 0.6 | **82.0 ± 2.9** |
| | **Gym Average** | 63.9 | 77.0 | 48.2 | 75.3 | 64.8 | 75.3 | 76.3 | **79.7** |
| Maze2D | maze2d-umaze | **94.7** | 42.1 | 65.7 | 14.8 | 16.4 ± 4.7 | 60.3 ± 7.7 | 73.2 ± 11.6 | 56.3 ± 4.1 |
| | maze2d-medium | 41.8 | 34.9 | 25.0 | 62.1 | 20.1 ± 8.4 | 37.0 ± 8.3 | 51.2 ± 4.9 | **136.8 ± 8.6** |
| | maze2d-large | 49.6 | 61.7 | 81.0 | **88.6** | 10.3 ± 8.6 | 25.4 ± 4.9 | 40.0 ± 10.2 | 84.4 ± 5.3 |
| | **Maze2D Average** | 62.0 | 46.2 | 57.2 | 55.2 | 15.6 | 40.9 | 54.8 | **92.5** |

and auto-bidding: (1) *Classical Offline RL Methods*, including BCQ (Fujimoto et al., 2019), CQL (Kumar et al., 2020), IQL (Kostrikov et al., 2022), TD3+BC (Fujimoto & Gu, 2021), and BEAR (Kumar et al., 2019). These traditional methods provide strong stability guarantees but offer limited expressiveness. (2) *Generative Sequence Modeling Methods*, including Behavior Cloning (BC), Decision Transformer (DT) (Chen et al., 2021), and PDiT (Mao et al., 2024). These methods are included to examine the limitations of deterministic or unimodal regression objectives, particularly the "Average Action" phenomenon. (3) *Auction-Specific Baselines*, consisting of constrained optimization methods such as CDT (Liu et al., 2023), as well as recent generative bidding models including GAS (Li et al., 2025), GAVE and GAVE-S (Gao et al., 2025), and the diffusion-based DiffBid (Guo et al., 2024), which represent advanced generative approaches for auto-bidding. Unless otherwise specified, DRIVE is implemented on a DT backbone.

**Evaluation Metrics.** On AuctionNet, the primary evalua-

tion metric in the main experiments is the *Value* $\sum r$, which measures unconstrained performance. For constrained bidding tasks, reported in Appendix C.5, we additionally adopt the *Score*, computed using the shaped reward $\sum r'$ defined in Equation (13), to explicitly reflect constraint satisfaction. On D4RL, the standard *Normalized Score* (Fu et al., 2020) is reported for fair comparison across tasks.

## 5.1. Overall Performance

This section presents a comprehensive comparison between DRIVE and baseline methods. Table 1 reports results on the AuctionNet benchmarks, while Table 2 summarizes normalized scores on the D4RL Gym and Maze2D domains. Additional results and extended analyses are provided in Appendix C.

**Results on AuctionNet.** Across all budget settings, DRIVE consistently achieves superior performance on AuctionNet. On the particularly challenging AuctionNet-Sparse bench-

*Table 3.* Ablation study on core components.

| Budget (%) | Actor Only (Dominant Mean) | Retr. + Critic (Selection) | Gen. + Critic (No Retr.) | DRIVE (Full) |
|---|---|---|---|---|
| 50 | $196.4 \pm 1.45$ | $185.4 \pm 1.86$ | $205.4 \pm 3.14$ | $\mathbf{211.8 \pm 1.57}$ |
| 75 | $290.1 \pm 3.32$ | $281.9 \pm 3.37$ | $296.7 \pm 2.39$ | $\mathbf{297.0 \pm 2.25}$ |
| 100 | $371.9 \pm 2.82$ | $370.4 \pm 1.58$ | $378.4 \pm 2.68$ | $\mathbf{399.0 \pm 3.74}$ |
| 125 | $450.3 \pm 3.85$ | $449.1 \pm 4.62$ | $461.3 \pm 6.70$ | $\mathbf{475.4 \pm 5.33}$ |
| 150 | $519.8 \pm 2.81$ | $490.3 \pm 1.10$ | $532.2 \pm 2.16$ | $\mathbf{550.6 \pm 4.64}$ |

*Table 4.* Effect of GMM vs Diffusion Head.

| Budget (%) | GMM Gen. + Critic (No Retr.) | Diffusion head+Critic |
|---|---|---|
| 50 | $\mathbf{205.4 \pm 3.14}$ | $190.7 \pm 2.21$ |
| 75 | $\mathbf{296.7 \pm 2.39}$ | $273.2 \pm 1.82$ |
| 100 | $\mathbf{378.4 \pm 2.68}$ | $376.3 \pm 5.72$ |
| 125 | $461.3 \pm 4.62$ | $\mathbf{467.2 \pm 2.53}$ |
| 150 | $\mathbf{532.2 \pm 2.16}$ | $531.4 \pm 4.05$ |

mark, purely generative baselines such as DT and DiffBid degrade noticeably due to severe data sparsity. In contrast, DRIVE effectively combines generative flexibility with value-based evaluation, leading to robust performance in low-density regimes. By anchoring action selection to retrieved high-quality decisions from the offline dataset, DRIVE mitigates the instability and hallucination issues commonly observed in purely parametric models. A detailed case study illustrating how DRIVE alleviates the Average Action issue by capturing multimodal bidding behaviors through the GMM-based policy is provided in Appendix C.

**Results on Gym Tasks.** As shown in the results, DRIVE attains the highest average normalized score across the Gym locomotion suite, outperforming both traditional value-based methods and recent generative approaches. While most methods exhibit similar performance on expert datasets, DRIVE shows clear advantages on mixed-quality datasets such as `medium` and `medium-replay`. For example, on `walker2d-medium-replay`, DRIVE outperforms DT by 14.7%. These results indicate that the proposed distributional action modeling combined with value-guided selection effectively alleviates the average-action issue in sequence modeling, enabling recovery of high-quality behaviors from suboptimal data.

**Results on Maze2D Domain.** In the Maze2D domain, DRIVE demonstrates a significant performance advantage over these baselines. On the challenging `maze2d-medium` task, DRIVE achieves a score of 136.8, exceeding all competing methods by a substantial margin. The strong performance on complex tasks highlights the importance of retrieval-augmented decision making. By grounding actions in retrieved transitions, DRIVE supports stable long-horizon planning and reduces instability in sparse regions. Furthermore, a qualitative analysis is presented in Appendix C.2.

### 5.2. Ablation Study

Table 3 reports a component-wise ablation analysis of the proposed framework. Comparing the deterministic *Actor Only* baseline with the *Gen. + Critic* variant reveals consistent performance gains across all budget settings, indicating that value-guided stochastic sampling is more effective than selecting a single most-likely action. By evaluating multiple sampled candidates, the critic is able to identify high-value actions that do not necessarily correspond to the dominant

mixture component, alleviating bias toward the most frequent bidding mode. Further incorporating the retrieval module (full DRIVE) leads to additional and substantial improvements. This demonstrates that retrieval-based augmentation provides an effective non-parametric correction by anchoring decision making to high-quality examples from the offline dataset. Such anchoring mitigates hallucinated or unreliable actions produced by the parametric policy, particularly near complex decision boundaries and under sparse data conditions.

### 5.3. Comparison of GMM and Diffusion Action Head

To validate the choice of the GMM-based action head, we perform a controlled experiment replacing it with a diffusion-based head (DDPM, T=100 steps) while keeping the Transformer backbone and IQL critic unchanged. Table 4 reports the average performance across budgets. GMM captures multimodal actions via a closed-form mixture likelihood, avoiding the "Average Action Trap" without the generative overhead of diffusion. Across all budgets, the GMM head achieves comparable or slightly better performance than the diffusion-based head. Importantly, it drastically reduces inference latency, requiring only 11 ms per step compared to 223 ms for the diffusion-based head. These results confirm that the GMM head provides a superior trade-off between performance and efficiency, making it well-suited for real-time bidding scenarios where low-latency decisions are critical. Overall, we chose the GMM head rather than diffusion to balance modeling parsimony, performance, and industrial feasibility.

### 5.4. Quantitative Analysis of Q-Function Multimodality

To evaluate how frequently Q-functions exhibit multimodality and its effect on the Average Action Trap, we randomly sample 2,000 states from the test set. For each state, we discretize the continuous action space into 100 uniformly spaced points and compute the corresponding Q-values. A state is classified as unimodal if its Q-function has a single peak, or multimodal if it has two or more local peaks. We evaluate the DT policy on these states by computing two metrics, the suboptimal rate, defined as the proportion of decision steps where the DT output action falls below the 80th percentile of Q-values among the 100 sampled actions, and the distance to the optimal action $a^* = \arg\max Q(s, a)$, measuring the absolute difference between DT's chosen

*Table 5.* Q-function multimodality analysis and DT suboptimality.

| Q Shape | Proportion | Suboptimal Rate (%) | Distancee to $a^*$ |
|---|---|---|---|
| Unimodal | 82.4% | 38.2 | 4.24 |
| Multimodal | 17.6% | 54.7 | 6.10 |
| Overall | 100% | 41.1 | 4.51 |

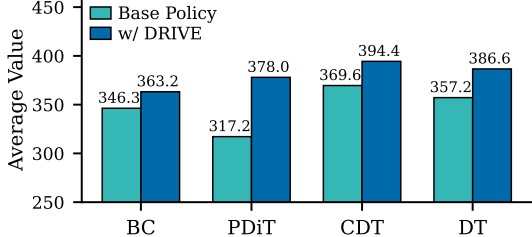

*Figure 4.* Performance across budget constraints for diverse backbone methods.

Each state is classified as unimodal (1 peak) or multimodal ($\geq$ 2 peaks) based on the number of local peaks in its Q-function.

Table 5 summarizes the overall statistics aggregated across all delivery periods P7–P13. We find that 17.6% of states are multimodal, where the DT mean-regression output falls into suboptimal regions with a suboptimal rate of 54.7%, compared to 38.2% for unimodal states. The average distance to the optimal action increases from 4.24 for unimodal states to 6.10 for multimodal states, confirming that the Average Action Trap is more severe in multimodal regions. This quantitative evidence supports the need for a GMM-based policy to preserve multiple bidding modes and mitigate systematic failure in offline auto-bidding. More detailed per-period visualizations and analysis are provided in Appendix C.3.

### 5.5. Generalization Across Diverse Backbones

To evaluate the general applicability of DRIVE, its core components are integrated into three representative Transformer-based policy backbones beyond the vanilla DT, including BC, CDT, and PDiT. As shown in Figure 4, this integration consistently improves performance across all backbones. In particular, the PDiT backbone achieves a substantial 19.2% improvement in average total reward. These results indicate that the proposed distributional and retrieval-augmented framework effectively mitigates mode collapse and sampling instability across different generative architectures, independent of the specific backbone design. Detailed numerical results are provided in Appendix C.4.

Further analyses are provided to validate the robustness of DRIVE under practical constraints. In particular, results in Appendix C.5 show that the constraint-aware critic is essential for constrained bidding tasks, as removing the penalty term leads to severe violation of CPA limits. Ap-

pendix C.6 compares different value-based critics, while sensitivity analyses in Appendix C.7 and Appendix C.8 demonstrate that DRIVE remains stable across a wide range of sampling and retrieval hyperparameters. Despite these performance gains, DRIVE incurs only a modest additional computational overhead at inference time. A detailed analysis of time and cost efficiency is provided in Appendix C.9.

## 6. Conclusion

This paper presents DRIVE, a unified framework for offline auto-bidding that addresses multimodal optimal behaviors and unreliable decision making under sparse data. DRIVE decouples candidate generation from decision-making and mitigates the "Average Action" trap by synergizing distributional GMM modeling with retrieval-augmented context. Extensive experiments on the industrial AuctionNet benchmark and standard D4RL tasks demonstrate that DRIVE significantly outperforms existing methods and highlights its exceptional generalization and transferability across different Transformer-based methods, providing a robust paradigm for deploying offline RL in bidding environments.

## Acknowledgements

This work was supported by the National Natural Science Foundation of China (Grant No.62506210).

## Impact Statement

This paper presents work whose goal is to advance the field of Machine Learning. There are many potential societal consequences of our work, none which we feel must be specifically highlighted here.

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

# A. Implementation Details

In this section, we provide a comprehensive account of the implementation details for the DRIVE framework to ensure reproducibility. We begin by summarizing the mathematical notation used throughout the paper. Subsequently, we present the complete algorithmic workflow, detailing the coordination between the GMM policy, retrieval mechanism, and offline critic. Finally, we specify the exact network architectures, retrieval engineering choices, and hyperparameter configurations used for both the AuctionNet and D4RL benchmarks.

## A.1. Table of Notation

*Table 6.* Table of Notation.

| Symbol | Explanation |
|---|---|
| $N$ | Total number of impression opportunities in a period. |
| $v_i, c_i$ | Value (e.g., conversion) and Cost of the $i$-th impression. |
| $B, \mathcal{K}_j$ | Total budget and target threshold for the $j$-th KPI. |
| $s_t, a_t, r_t$ | State, Action, and Reward at timestep $t$. |
| $\hat{R}_t$ | Return-to-go (RTG) conditioned at timestep $t$. |
| $\mathcal{D}$ | Offline dataset of trajectories $\tau$. |
| $\tau_{0:t-1}$ | Trajectory prefix from timestep 0 to $t-1$. |
| $M$ | Number of mixture components in the GMM head. |
| $\alpha_m, \mu_m, \sigma_m^2$ | Mixing coef., mean, and variance of the $m$-th component. |
| $\mathcal{N}(\cdot \mid \mu_m, \sigma_m^2)$ | $m$-th Gaussian density in the mixture. |
| $\mathcal{L}_{\text{GMM}}$ | GMM log-likelihood training objective. |
| $L$ | Number of generated candidate actions from the GMM distribution of action. |
| $\mathcal{A}_{\text{gen}}$ | Set of candidate actions generated from the GMM policy. |
| $\mathcal{I}$ | Retrieval index built from offline embeddings. |
| $h_t$ | Contextual state embedding at timestep $t$. |
| $f_{\text{enc}}$ | Transformer encoder that produces $h_t$. |
| $d$ | Dimensionality of contextual state embeddings $h_t$. |
| $K_{\text{pool}}$ | Size of the initial retrieved candidate pool. |
| $\mathcal{C}_{\text{pool}}$ | Retrieved pool with actions and RTG statistics. |
| $K$ | Number of retrieved actions kept after RTG-based filtering. |
| $\mathcal{A}_{\text{ret}}$ | Set of retrieved candidate actions. |
| $\mathcal{A}_{\text{cand}}$ | Unified candidate pool $\mathcal{A}_{\text{gen}} \cup \mathcal{A}_{\text{ret}}$. |
| $Q_i(s, a)$ | $i$-th Q-function used in the critic (IQL). |
| $V(s)$ | State value function in IQL. |
| $\mathcal{L}_V$ | Expectile regression loss for learning $V(s)$. |
| $\mathcal{L}_Q$ | Bellman regression loss for learning $Q(s, a)$. |
| $L_2^{\eta}(\cdot)$ | Asymmetric squared loss for expectile regression. |
| $\eta$ | Expectile level controlling implicit maximization ($0.5 < \eta < 1$). |
| $\gamma$ | Discount factor in value learning. |
| $a^*$ | Final selected action at inference time. |

## A.2. Algorithm Description

The complete workflow of our proposed framework, DRIVE, is summarized in Algorithm 1. The procedure begins by training the GMM-based policy $\pi_\theta$ and the value-based critic $Q_\phi$ on the offline dataset $\mathcal{D}$. Subsequently, a retrieval index $\mathcal{I}$ is constructed using representative state embeddings. Finally, the inference process integrates generative sampling, RTG-guided retrieval, and conservative value evaluation to select the optimal action $a^*$.

## A.3. Retrieval Implementation Details

We facilitate efficient and effective retrieval through a rigorous filtering strategy and a high-performance approximate nearest neighbor search implementation.

**Index Construction and Filtering.** To ensure the retrieval mechanism provides high-quality guidance, we implement a lightweight filtering step during index construction. Specifically, we exclude transitions associated with a zero Return-to-Go, where $\hat{R}_t = 0$. These instances typically correspond to unsuccessful bids or impressions that yielded no value, rendering them uninformative for policy improvement. Filtering these samples not only improves the relevance of retrieved candidates

---

**Algorithm 1** The DRIVE Framework

---

**Require:** Offline dataset $\mathcal{D}$; Number of GMM components $M$; Target return $\hat{R}_t$;
**Require:** Hyperparameters: $L$ (gen samples), $K_{\text{pool}}$ (sim pool), $K$ (retrieval final)
 1: **// Phase 1: Policy Training**
 2: Initialize Policy $\pi_\theta$
 3: **while** policy not converged **do**
 4:     Sample batch of trajectories $\tau \sim \mathcal{D}$
 5:     Update Policy $\pi_\theta$ by minimizing $\mathcal{L}_{\text{GMM}}$ (Eq. 6)
 6: **end while**
 7: **// Phase 2: Retrieval Index Construction**
 8: Initialize $\mathcal{I} \leftarrow \emptyset$
 9: **for** $\tau \in \mathcal{D}$ **do**
10:     Compute state embeddings $h_t$ (Eq. 8)
11:     Store $(h_t \rightarrow \{a_t, \hat{R}_t\})$ into $\mathcal{I}$
12: **end for**
13: **// Phase 3: Critic Training**
14: Initialize Critic $Q_\phi, V_\psi$
15: **while** critic not converged **do**
16:     Sample transitions $(s, a, r, s') \sim \mathcal{D}$
17:     Update $V_\psi$ via expectile regression (Eq. 11)
18:     Update $Q_\phi$ via Bellman error (Eq. 12)
19: **end while**
20: **Evaluation:**
21: **for** each decision step $t$ **do**
22:     **Encoding:** Get $h_t$ and GMM params via $\pi_\theta(\tau_{0:t-1}, \hat{R}_t, s_t)$
23:     **Generation:** $\mathcal{A}_{\text{gen}} \leftarrow \{a^{(l)} \sim \sum \alpha_m \mathcal{N}(\mu_m, \sigma_m^2)\}_{l=1}^{L}$
24:     **Retrieval:**
25:     Query neighbors: $\mathcal{C}_{\text{pool}} \leftarrow \text{Top-}K_{\text{pool}}^{\text{sim}}(\mathcal{I}, h_t)$
26:     Filter by RTG: $\mathcal{A}_{\text{ret}} \leftarrow \text{Top-}K^{\hat{R}}(\mathcal{C}_{\text{pool}})$
27:     **Execution Phase:**
28:     Pool: $\mathcal{A}_{\text{cand}} \leftarrow \mathcal{A}_{\text{gen}} \cup \mathcal{A}_{\text{ret}}$
29:     Select: optimal action $a^*$ (Eq. 14)
30:     Execute $a^*$
31: **end for**

---

but also reduces the memory footprint of the index.

**Scalable Implementation.** To ensure scalability and low-latency inference, particularly for large-scale trajectory datasets, we implement the retrieval mechanism using the FAISS library (Johnson et al., 2019). Given the contextual embeddings $h_t$ produced by the Transformer encoder, we construct the retrieval index using the Hierarchical Navigable Small World (HNSW) algorithm. Specifically, the inner product metric is employed for similarity calculation. Since all embeddings are pre-normalized, this effectively performs Cosine Similarity search. The HNSW graph structure allows for logarithmic-time complexity during queries, offering a superior trade-off between search speed and retrieval recall compared to exact search methods. To further enhance robustness during inference, we employ an oversampling strategy where $3 \times K$ nearest neighbors are initially retrieved based on cosine similarity. From this expanded pool, we filter out invalid entries and select the final $K$ candidates with the highest stored RTG values.

### A.4. Hyperparameters Setting

We summarize the comprehensive hyperparameter settings and architectural details for all components in Table 7.

**AuctionNet Configuration.** We adopt a dual-model strategy for the AuctionNet environment to balance decision-making capacity with retrieval efficiency. The main policy network is configured with a high-capacity backbone to capture the complex, multimodal decision boundaries inherent in auto-bidding.In contrast, the auxiliary encoder is designed as a

lightweight architecture with a reduced embedding dimension. This compression is critical for minimizing the storage footprint of the vector index and accelerating similarity search. While the main network uses a sliding window ($K = 20$) for efficient inference, we set the context length of the retrieval encoder to 48, matching the full episode length of the AuctionNet environment. This design allows the encoder to attend to the complete history of an advertising period, generating a global and comprehensive trajectory representation for effective indexing.

**D4RL Configuration.** For the D4RL benchmarks, we strictly adhere to the standard Decision Transformer architecture to ensure a fair comparison with baseline methods. Unlike the AuctionNet setup, the latent space in Gym tasks is sufficiently compact for direct vector search. Consequently, as noted in the table, we do not employ a separate retrieval encoder for these tasks. Instead, we directly utilize the contextualized representations output by the policy backbone as query vectors.

*Table 7.* Hyperparameter and Architecture Comparison. Summary of the model configurations for the main AuctionNet policy, the auxiliary encoder used for retrieval, and the model used for Gym locomotion tasks.

| Configuration | AuctionNet (Policy) | AuctionNet (Encoder) | D4RL(No Separate Encoder) |
|---|---|---|---|
| Activation Function | ReLU | ReLU | GELU |
| Embedding Dim ($d$) | 512 | 64 | 128 |
| Number of Layers | 6 | 3 | 3 |
| Attention Heads | 8 | 4 | 1 |
| Dropout | 0.1 | 0.1 | 0.1 |
| Context Length ($K$) | 20 | 48 | 20 |
| Max Episode Length | 48 | 48 | 1000 |
| Optimizer | AdamW | AdamW | AdamW |
| Learning Rate | $1 \times 10^{-5}$ | $1 \times 10^{-3}$ | $1 \times 10^{-4}$ |
| Weight Decay | $1 \times 10^{-4}$ | $1 \times 10^{-3}$ | $1 \times 10^{-4}$ |
| Warmup Steps | 10,000 | 10,000 | 10,000 |
| Batch Size | 256 | 256 | 64 |

**Critic Implementation Details.** We detail the specific hyperparameter configurations and architectural choices for the IQL critic in Table 8. Unlike the standard wide MLP architectures typically employed in D4RL continuous control benchmarks, we adopt a more compact network structure optimized for the AuctionNet environment. Specifically, the V-network uses a tapered structure (decreasing hidden sizes) to efficiently process the auction state features. The optimization hyperparameters, including learning rates and soft update frequencies, were also calibrated to ensure training stability and convergence in this domain.

*Table 8.* Critic Hyperparameter Comparison between AuctionNet and D4RL Benchmarks. Summary of architecture and optimization details for IQL (Critic) modules.

| Hyperparameter | AuctionNet (Ours) | D4RL Benchmarks |
|---|---|---|
| Q-Network Hidden Sizes | [64, 64] | [256, 256] |
| V-Network Hidden Sizes | [128, 64, 32] | [256, 256] |
| Activation Function | ReLU | ReLU |
| Optimizer | Adam | Adam |
| Critic Learning Rate | $1 \times 10^{-4}$ | $3 \times 10^{-4}$ |
| Soft Update Rate ($\tau$) | 0.01 | 0.005 |
| Expectile ($\eta$) | 0.7 | 0.7 |
| Discount Factor ($\gamma$) | 0.99 | 0.99 |

# B. Experiment Details

In this section, we detail the experimental setups used to evaluate the proposed DRIVE framework. We primarily utilize the industrial **AuctionNet** dataset to assess performance in realistic auto-bidding scenarios, with a specific focus on the challenges posed by sparse rewards and long-tail action distributions. Additionally, to demonstrate the method's universality beyond bidding, we extend our evaluation to the standard **D4RL** continuous control benchmarks. The specifications, state

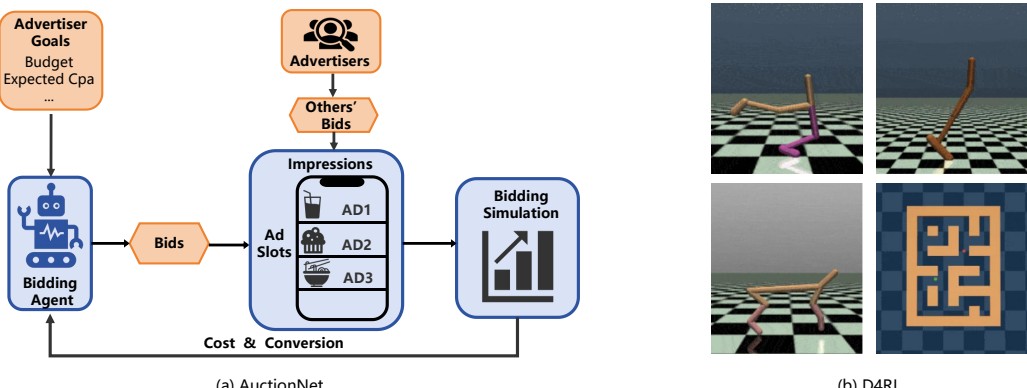

(a) AuctionNet.
(b) D4RL.

*Figure 5.* Visualizations of the AuctionNet and D4RL benchmarks.

features, and distributional characteristics of these environments are described below.

### B.1. Dataset Details

We primarily evaluate DRIVE on the AuctionNet dataset for auction bidding. To assess its generalization capabilities, we further extend our evaluation to the D4RL benchmark for continuous control. Visualizations of these tasks are provided in Figure 5.

**AuctionNet.** AuctionNet has two versions, with AuctionNet-sparse characterized by sparse rewards. Each dataset is organized into multiple delivery periods. Every period encompasses approximately 500,000 impression opportunities and is temporally segmented into 48 decision steps. Detailed parameters are shown in Table 9.

*Table 9.* The Parameters of AuctionNet and AuctionNet-sparse.

| Params | AuctionNet | AuctionNet-Sparse |
| --- | --- | --- |
| Trajectories | 479,376 | 479,376 |
| Delivery Periods | 9,987 | 9,987 |
| Time steps in a trajectory | 48 | 48 |
| State dimension | 16 | 16 |
| Action dimension | 1 | 1 |
| Return-To-Go Dimension | 1 | 1 |
| Action range | [0, 493] | [0, 589] |
| Impression's value range | [0, 1] | [0, 1] |
| CPA range | [6, 12] | [60, 130] |
| Total conversion range | [0, 1512] | [0, 57] |

The trajectory-formatted data is aggregated from the raw traffic logs and captures the decision-making process by recording the information for multiple advertisers at every step. The detailed state is provided below:

- time_left: Represents the remaining time in the current delivery period.
- budget_left: Indicates the advertiser's remaining budget available for the current period.
- historical_bid_mean: The average bid price placed by the advertiser across all preceding time steps.
- last_three_bid_mean: The moving average of the advertiser's bid prices over the most recent three time steps.
- historical_LeastWinningCost_mean: The historical average of the market price (minimum cost to win an impression) observed in previous steps.
- historical_pValues_mean: The average conversion probability (p-value) of impressions in the past time steps.
- historical_conversion_mean: The average number of conversion events achieved by the advertiser in prior steps.
- historical_xi_mean: The historical winning rate, calculated as the average binary winning status, where 1 represents winning the impression and 0 represents not winning.
- last_three_LeastWinningCost_mean: The average of the least winning costs over the last three time steps.
- last_three_pValues_mean: The average conversion probability of impressions over the last three time steps.

- last_three_conversion_mean: The average number of conversions obtained during the last three time steps.
- last_three_xi_mean: The recent winning rate, representing the average winning status over the last three time steps.
- current_pValues_mean: The average conversion probability of all impression opportunities in the current time step.
- current_pv_num: The total volume of impression opportunities available at the current time step.
- last_three_pv_num_total: The cumulative number of impression opportunities served over the last three time steps.
- historical_pv_num_total: The total accumulated count of impression opportunities over past time steps.

The experiments are performed in a simulation environment resembling a real-world commercial advertising system (Su et al., 2024). An episode represents one delivery day, segmented into 48 time steps, with a traffic volume of approximately 500,000 impressions. The competition landscape consists of 48 advertisers with varying budgets and CPA constraints. In the evaluation phase, our model controls a target advertiser. To rigorously evaluate performance robustness, we conduct repeated trials using different advertiser profiles and delivery periods, taking the average value as the final evaluation score.

To comprehensively evaluate the complexity shift between the dense and sparse settings, we visualize the action distributions of both the original AuctionNet and the AuctionNet-Sparse datasets. As illustrated in Figure 6, the comparison reveals fundamental structural differences that underscore the difficulty of the sparse control task.

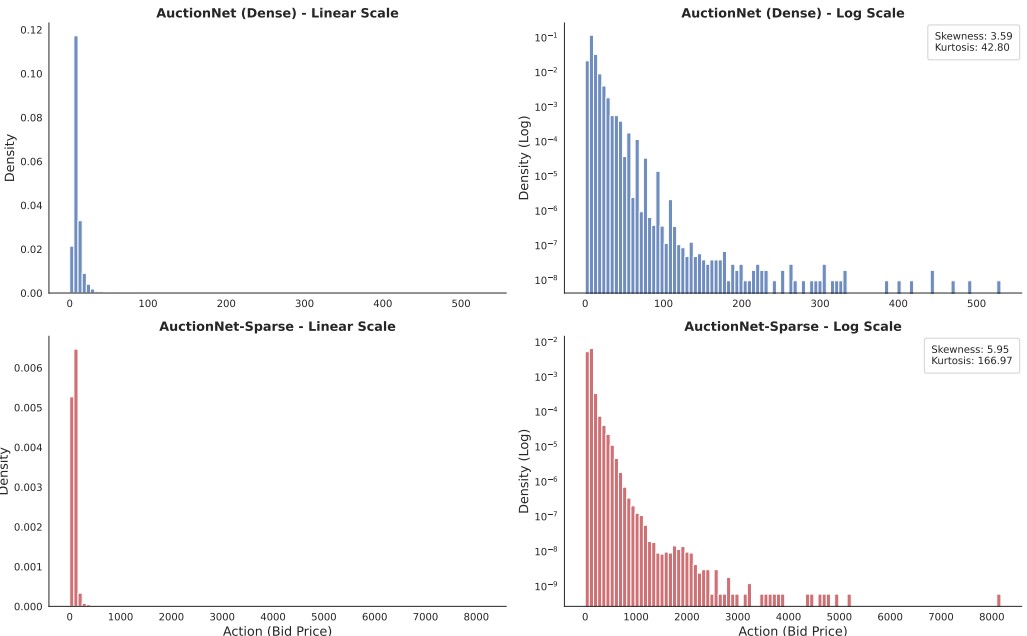

*Figure 6.* Comparison of Action Distributions: AuctionNet vs. AuctionNet-Sparse. The top row shows the original dense dataset, while the bottom row depicts the sparse variant. The log-scale plots (right column) reveal the extreme long-tail property of the sparse dataset, where the action space extends significantly with a drastic increase in kurtosis. This shift highlights the exploration difficulty in the sparse setting.

While the dense dataset concentrates actions within a relatively compact range, the sparse variant exhibits a significantly broader action coverage. The linear-scale plots (left column) demonstrate that although the majority of bidding actions remain in the lower value region for both datasets, the sparse setting requires the agent to generalize over a much wider and sparser manifold. The logarithmic-scale plots (right column) highlight the heavy-tailed nature of the distributions. The sparse dataset demonstrates a more pronounced long-tail property compared to the dense baseline. This structural shift implies that the agent must handle a higher degree of distributional shift and learn to retrieve valid high-value actions that are statistically rare but critical for optimal performance. These observations justify the necessity of employing robust generative baselines capable of modeling multi-modal distributions and handling extreme outliers, rather than relying on simple unimodal regression objectives.

**D4RL.** D4RL (Datasets for Deep Data-Driven Reinforcement Learning) serves as a standardized benchmark suite for offline reinforcement learning. In the domain of high-dimensional continuous control, the Gym-MuJoCo tasks simulate articulated robots with varying degrees of freedom: `hopper` (a monoped), `walker2d` (a planar biped), and `halfcheetah` (a two-legged cheetah-like robot). The objective in these environments is to master stable locomotion behaviors to maximize

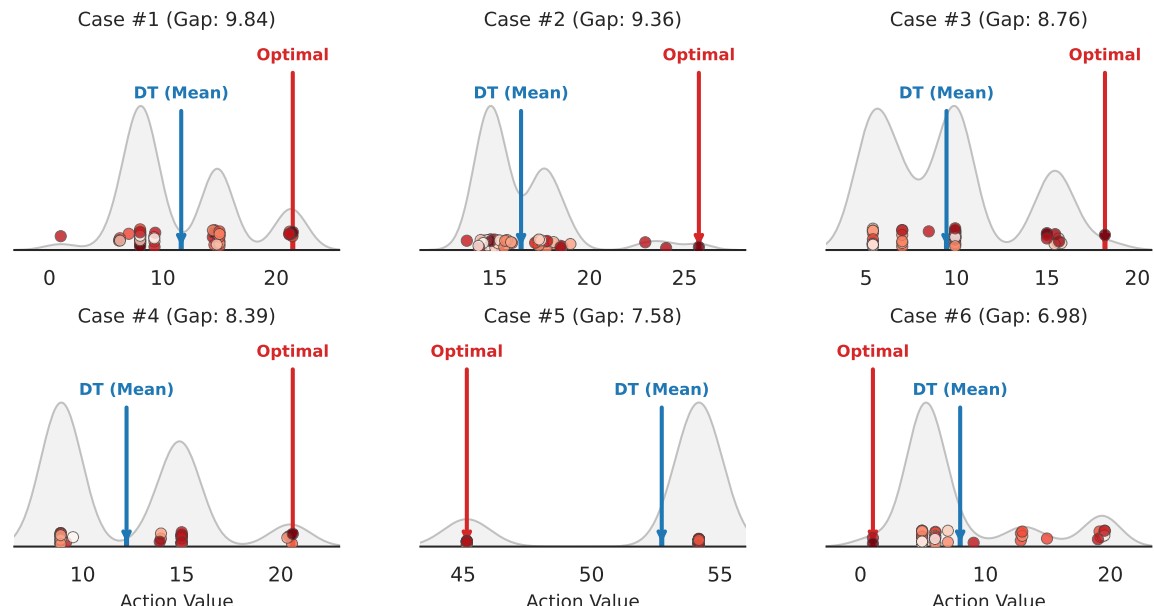

*Figure 7.* Representative retrieval neighborhoods illustrating the discrepancy between the DT prediction and the optimal action. Each panel shows the empirical distribution of actions, the mean action predicted by the DT head (blue vertical line, "DT (Mean)"), and the action that achieves the highest return in that neighborhood (red vertical line, "Optimal"). "Gap" denotes the absolute difference between these two actions.

forward velocity while minimizing control costs. Complementing these, the Maze2D domain focuses on sparse-reward navigation, requiring a 2D agent to traverse complex geometric layouts to reach specified target coordinates. The benchmark provides datasets of diverse qualities to test the algorithm's generalization.

**Applicability of GMM to General Offline RL.** Although DRIVE is motivated by auto-bidding, the challenges it addresses, multimodal action distributions and data heterogeneity, are ubiquitous in general offline RL benchmarks. In Gym-MuJoCo tasks, particularly in suboptimal datasets like `medium-replay`, the data consists of a mixture of high-performing expert behaviors and low-quality exploratory noise. A unimodal policy risks averaging these conflicting behaviors into a mediocre action. DRIVE's distributional modeling, combined with critic-guided selection, allows the agent to disentangle optimal behaviors from noisy data, effectively recovering high-reward actions from mixed-quality distributions. In Maze2D tasks, the optimal policy is inherently multimodal, such as bypassing obstacles from either left or right. Standard deterministic regressors tend to output invalid interpolated actions like crashing into the wall, whereas our GMM head explicitly captures these distinct valid trajectories.

# C. Additional Results

## C.1. Visualization of Multi-Modal Action Distributions

To provide a concrete intuition for the limitations of deterministic policies in offline RL, we select and visualize specific instances where the retrieved neighborhood exhibits complex, multi-modal structures. Figure 7 presents six representative examples identified from the evaluation set. These cases were chosen to explicitly illustrate the "Average Action" Trap, where the distribution of candidate actions contains distinct modes rather than a single unimodal cluster.

In these visualizations, the gray curves represent the empirical distribution of actions found in the retrieved history. The blue vertical lines denote the expected action predicted by a standard Decision Transformer head (trained via MSE), which mathematically converges to the conditional mean of the distribution. Crucially, we observe that the return-maximizing actions (red vertical lines) consistently reside within the high-density modes. We define the *gap* as

$$\text{Gap} = |a_{\text{DT mean}} - a_{\text{optimal}}|,$$

The significant Gap between the mean prediction and the optimal action highlights the failure mode of deterministic heads, which average over conflicting strategies, resulting in falling into regions corresponding to suboptimal behaviors.

This qualitative evidence underscores the necessity of the GMM policy head used in DRIVE, which is designed to capture these distinct modes explicitly rather than collapsing them into a single mean.

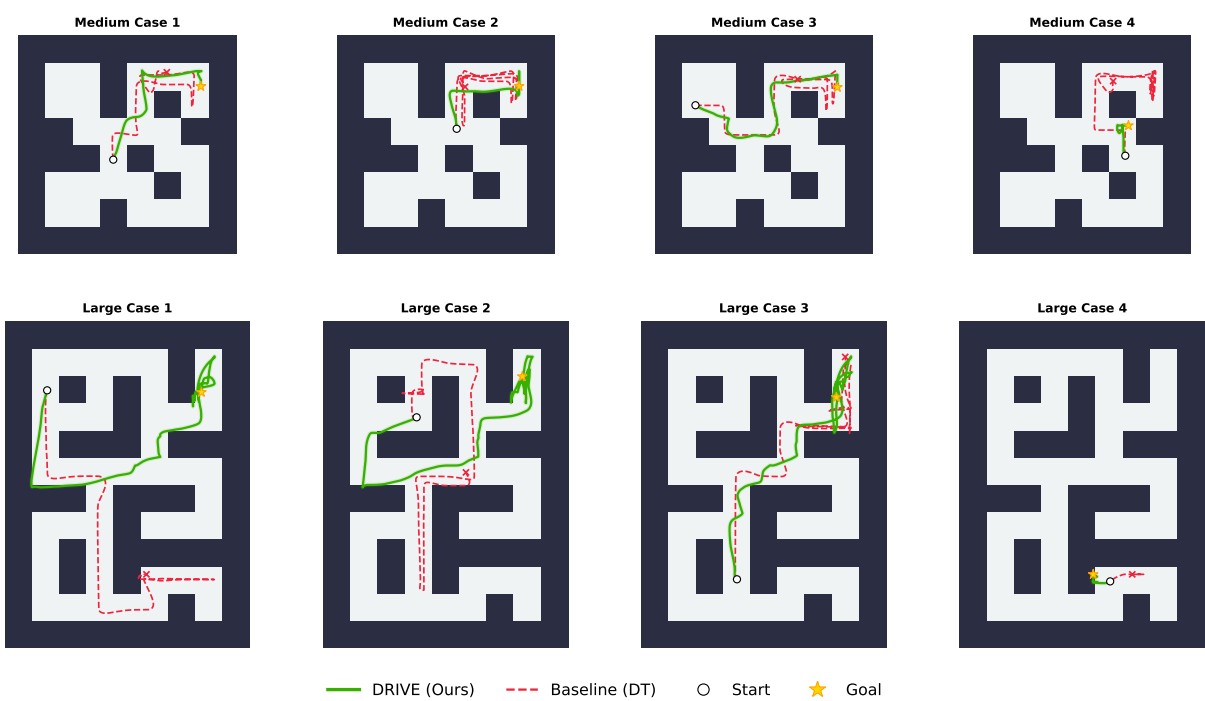

*Figure 8.* Qualitative comparison of trajectories on Maze2D. The top row displays results on `maze2d-medium`, and the bottom row on `maze2d-large`. Green solid lines denote DRIVE (Ours), and red dashed lines denote the DT baseline. DRIVE consistently generates collision-free paths to the goal without manual seed selection.

## C.2. Visualization on Maze2D

In addition to the quantitative evaluations presented in the main text, Figure 8 presents a qualitative analysis of the learned policies via rollout trajectories on the `maze2d-medium` and `maze2d-large` tasks. To ensure an unbiased comparison, manual curation of successful cases is strictly avoided. Instead, four random seeds are uniformly sampled for each environment variant. For each seed, the environment is initialized with fixed start and goal positions, followed by the execution of a single evaluation episode for both the DRIVE agent and the DT baseline. All eight resulting trajectory pairs are visualized directly without selection. As illustrated, the DRIVE agent consistently plans shorter and more stable paths to navigate around obstacles, successfully reaching the goal in all sampled scenarios. In contrast, the DT baseline frequently fails to find feasible paths, resulting in collisions with walls or stagnation in suboptimal regions. These visualizations qualitatively corroborate the significant performance improvements and superior long-horizon planning capabilities of DRIVE observed in the quantitative results.

## C.3. Q-Function Multimodality Analysis

To provide a more detailed view of Q-function multimodality, we analyze a per-period breakdown across delivery periods P7–P13. Figure 9 illustrates the per-period results, providing a detailed view of multimodal prevalence, DT suboptimal rates, and distances to the optimal action. Overall, multimodal states consistently constitute a notable fraction of the state space and are systematically more challenging: their DT suboptimal rates are higher, and the average distance to the optimal action is larger compared to unimodal states. These trends persist across all delivery periods, illustrating that the severity of the Average Action Trap is generally consistent over time. This analysis further motivates the use of a GMM-based policy to preserve multiple bidding modes and mitigate systematic failures in offline auto-bidding.

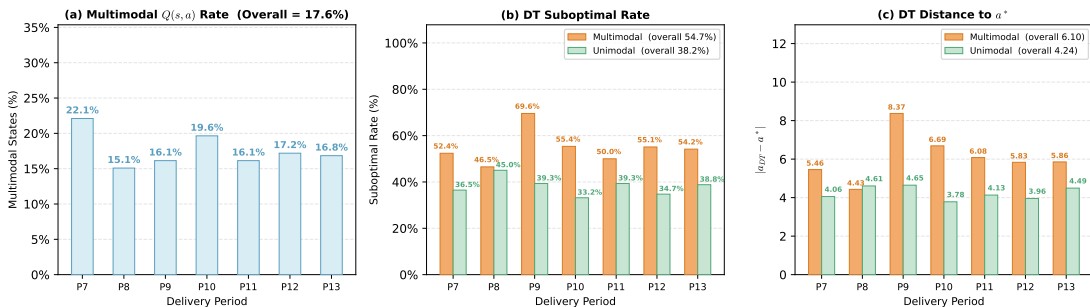

*Figure 9.* Q-function multimodality analysis across delivery periods P7–P13 (2,000 randomly sampled states). (a) Proportion of states with multimodal Q-functions ($\geq 2$ peaks). (b) DT suboptimal rate (Q-percentile $< 80\%$) for multimodal vs. unimodal states. (c) Average DT distance to $a^*$ for multimodal vs. unimodal states.

### C.4. Generalization across DT-style Architectures

While our primary analysis centers on the standard Decision Transformer, we contend that the "Average Action" issue is a systemic limitation inherent to the entire family of sequence modeling policies. To validate the generalization capabilities of DRIVE across this broader class, we incorporated it into three representative Transformer-based baselines, the original DT (Chen et al., 2021), CDT (Liu et al., 2023), and PDiT (Mao et al., 2024). Additionally, we include Behavior Cloning (BC) to serve as a fundamental regression benchmark.

Table 10 presents the detailed numerical comparison on the AuctionNet dataset. Despite their architectural differences, all DT-style baselines suffer from mode collapse in multimodal auction landscapes. By augmenting them with DRIVE's distributional head and retrieval mechanism, we observe consistent and significant performance gains across all budget settings. Most notably, the PDiT backbone achieves the largest improvement, confirming that our framework effectively complements sophisticated sequence models by providing explicit, high-quality historical anchors to rectify generative hallucinations.

*Table 10.* Evaluation of Generalization across Architectures. The DRIVE module is integrated into four distinct policy backbones (BC, PDIT, CDT, and DT) on the AuctionNet dataset. We report the total rewards (mean $\pm$ standard deviation) over 5 seeds.

| Dataset | Budget | BC | BC+DRIVE | PDIT | PDIT+DRIVE | CDT | CDT+DRIVE | DT | DT+DRIVE(ours) |
|---|---|---|---|---|---|---|---|---|---|
| | 50% | $201 \pm 1.30$ | $\mathbf{208 \pm 1.48}$ | $187 \pm 1.14$ | $\mathbf{207 \pm 0.79}$ | $208 \pm 2.06$ | $\mathbf{218 \pm 2.18}$ | $208 \pm 1.75$ | $\mathbf{212 \pm 1.57}$ |
| | 75% | $283 \pm 1.18$ | $\mathbf{287 \pm 2.64}$ | $267 \pm 4.47$ | $\mathbf{300 \pm 2.00}$ | $300 \pm 2.00$ | $\mathbf{304 \pm 2.03}$ | $298 \pm 2.21$ | $\mathbf{297 \pm 2.25}$ |
| AuctionNet | 100% | $358 \pm 2.72$ | $\mathbf{371 \pm 3.13}$ | $328 \pm 3.51$ | $\mathbf{387 \pm 4.94}$ | $382 \pm 3.19$ | $\mathbf{409 \pm 2.44}$ | $373 \pm 3.18$ | $\mathbf{399 \pm 3.74}$ |
| | 125% | $419 \pm 3.38$ | $\mathbf{440 \pm 4.44}$ | $381 \pm 1.29$ | $\mathbf{464 \pm 0.17}$ | $450 \pm 2.68$ | $\mathbf{488 \pm 3.54}$ | $430 \pm 2.98$ | $\mathbf{475 \pm 5.33}$ |
| | 150% | $471 \pm 5.11$ | $\mathbf{510 \pm 2.68}$ | $423 \pm 2.56$ | $\mathbf{532 \pm 4.94}$ | $508 \pm 2.67$ | $\mathbf{553 \pm 3.77}$ | $477 \pm 2.12$ | $\mathbf{551 \pm 4.64}$ |
| Avergae | | 346.3 | $\mathbf{363.2}$ ↑16.9 | 317.2 | $\mathbf{378.0}$ ↑60.8 | 369.6 | $\mathbf{394.4}$ ↑24.8 | 357.2 | $\mathbf{386.6}$ ↑29.4 |

### C.5. Effect of Constraint-Aware Training.

We examine the necessity of aligning the critic's objective with the specific constraints of the environment. Since the AuctionNet task imposes a CPA limit, we incorporate a corresponding penalty term into the critic and evaluate performance by analyzing the trade-off between the "Overall Score" and constraint compliance. In unconstrained settings, the critic would simply optimize the raw reward. However, as shown in Figure 10, a naive critic trained solely on raw conversions fails to internalize the cost of violations, leading to a high CPA Exceed Rate (red line). In contrast, by injecting the constraint-specific penalty into the IQL objective, our method effectively shapes the value landscape to reflect the true task goal. The critic learns to assign lower values to high-CPA actions, effectively filtering out unsafe candidates during the selection phase. This confirms that for constrained tasks, the value estimation can be explicitly tailored to the active constraints to ensure safety and optimality.

### C.6. Impact of Critic Architecture.

To justify the choice of the critic module in DRIVE, we conduct an ablation study comparing our IQL-based critic with a CQL-based critic. Figure 11 illustrates the total rewards under varying budget constraints.

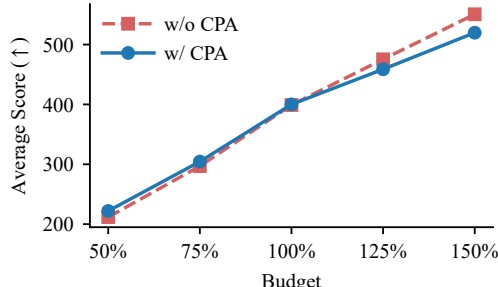 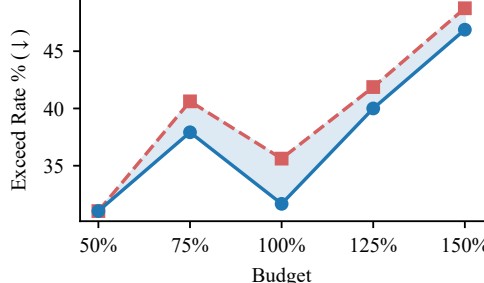

*Figure 10.* Detailed Performance Analysis across Budgets. Comparison of Average Score (Left) and Risk/Exceed Rate (Right) under varying budget constraints. The constraint-aware critic (w/ CPA) consistently maintains high rewards while significantly reducing the risk of budget violation compared to the baseline.

Both critics yield comparable performance. The conservative nature of CQL is effective when the budget is tight, as avoiding overestimation is crucial for preventing budget depletion. The IQL-based critic significantly outperforms the CQL variant. As the budget increases, the agent requires a more accurate estimation of the upper quantiles of the return distribution to identify high-value opportunities. IQL, which performs expectile regression, avoids the under-estimation bias often observed in CQL, thereby providing better guidance for the retrieval and ranking process. Based on these results, we adopt IQL as the default value estimator to ensure robust performance across all budget levels.

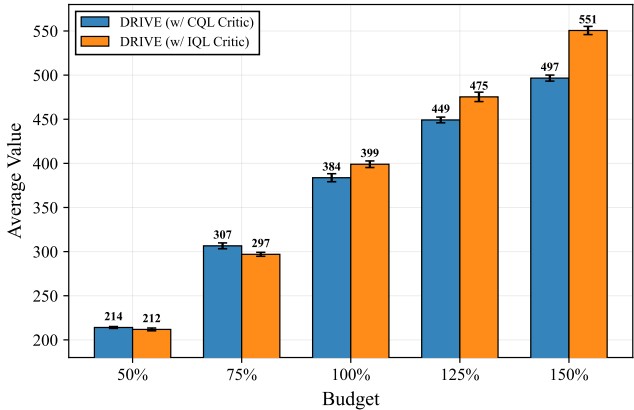

*Figure 11.* Ablation study on Critic architecture. Performance comparison between the CQL-based critic (Blue) and the IQL-based critic (Orange) across varying budget constraints. The results indicate that while both architectures yield comparable performance in low-budget regimes, the IQL critic demonstrates superior scalability and robustness, consistently outperforming the CQL variant as the budget increases.

### C.7. Sensitivity to Sampling Size $L$.

We investigate the impact of the number of sampled actions $L$ on performance, as shown in Figure 12. Overall, increasing $L$ from 1 to 32 does not result in monotonic improvements for both variants. The performance gains from additional samples are marginal, and the curves remain relatively flat. This indicates that the GMM-based policy already produces reasonably good candidates even with a small sampling size. Across all sampling sizes, the retrieval-augmented variant consistently achieves higher average reward and exhibits smaller fluctuations than the pure generative baseline. These results suggest that, while the overall method is not highly sensitive to the exact choice of $L$, retrieval provides complementary high-quality candidates that stabilize performance and reduce the reliance on extensive sampling from the generative policy.

### C.8. Sensitivity to Number of Retrieved Neighbors $K$.

We investigate the impact of the number of retrieved neighbors $K$ on performance, as visualized in Figure 13. Unlike the sampling size analysis, the "only retrieval" baseline exhibits a strong sensitivity to $K$, showing monotonic performance improvements as $K$ increases from 1 to 9. This confirms that purely retrieval-based approaches rely heavily on a larger candidate pool to ensure the coverage of high-quality actions. In contrast, DRIVE demonstrates remarkable robustness, maintaining consistently superior performance across all tested $K$ values. Notably, our method achieves near-optimal

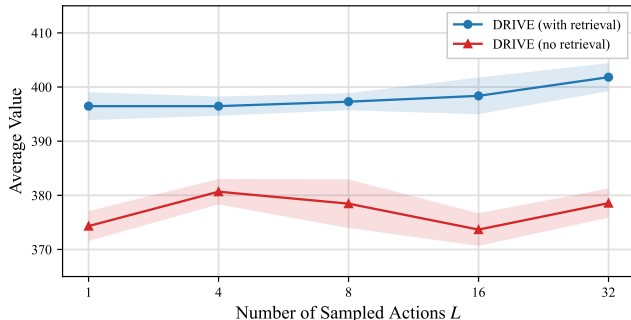

*Figure 12.* Sensitivity analysis of sampling size $L$. Average Total Reward as the number of generated candidate actions $L$ increases. Error bars denote the standard deviation over 5 seeds. While the performance is robust to variations in $L$, the retrieval-augmented variant consistently outperforms the baseline across all sampling budgets.

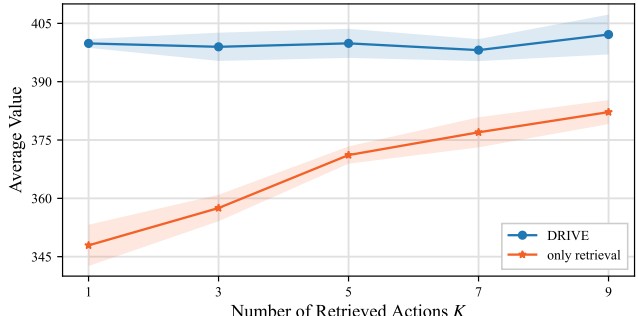

*Figure 13.* Sensitivity analysis of retrieved neighbor count $K$. Average Total Reward comparison between DRIVE and the retrieval-only baseline. Error bars denote standard deviation over 5 seeds. While only retrieve relies on a larger $K$ for performance, DRIVE demonstrates superior robustness, maintaining high rewards even with minimal retrieval context ($K = 1$).

rewards even with a minimal context of $K = 1$, significantly surpassing the baseline's peak performance at $K = 9$. These results indicate that the GMM-based policy effectively extracts and generalizes from sparse retrieval signals, decoupling the system's effectiveness from the strict quantity of retrieved neighbors and ensuring reliability even under constrained retrieval budgets.

### C.9. Computational Efficiency Analysis

In real-time bidding (RTB) systems, strict latency constraints (typically $< 50$ms or $< 100$ms) are imposed to ensure timely ad delivery. Therefore, it is crucial to verify that the performance gains of DRIVE do not come at the cost of prohibitive computational overhead. We analyze the computational cost from both training and inference perspectives.

**Training Cost.** The training cost of DRIVE remains comparable to standard baseline methods. The GMM-based policy head introduces only a negligible increase in parameters compared to the Transformer backbone. Furthermore, the construction of the retrieval index and the training of the offline critic are performed entirely offline. Consequently, they do not impose any additional burden on the online training loop or real-time infrastructure.

**Inference Latency.** To evaluate the real-time feasibility of DRIVE, we conducted a rigorous latency comparison against the standard Decision Transformer baseline. We measured the average wall-clock time per decision step over three independent runs. The DT baseline achieves an average latency of $10.44$ ms. DRIVE, which incorporates distributional sampling, retrieval, and value evaluation, records an average latency of $46.38$ ms. Although DRIVE introduces additional latency compared to the vanilla DT, the breakdown indicates that the combined time for candidate generation (including GMM sampling and retrieval) averages $39.01$ ms, while the critic evaluation takes only $7.37$ ms. Crucially, the total inference time consistently remains below $50$ ms. This demonstrates that DRIVE strikes a favorable trade-off, delivering significant performance improvements while satisfying the strict latency requirements of industrial RTB systems.

**Memory Usage.** We conducted a comparative analysis of memory consumption between DRIVE and the DT baseline. The standard DT model requires a peak CPU RAM of 9.36 GB, which represents the base cost for data loading and environment simulation. In contrast, DRIVE records a peak CPU RAM of 28.94 GB. A detailed breakdown reveals that the majority of

this overhead is explicitly allocated to the retrieval mechanism. Specifically, the pre-built FAISS index accounts for 13.33 GB (approx. 68% of the incremental memory). The remaining increase is attributed to auxiliary structures (e.g., cached RTG values and action candidates) and runtime buffers required for the retrieval-augmented inference. Given that modern industrial inference servers typically possess 64GB to 256GB of RAM, this memory footprint is well within acceptable limits for deployment.

