# OpenReview forum: "DRIVE: Distributional and Retrieval-Augmented Bidding with Value Evaluation"
_ICML.cc/2026/Conference — ICML 2026 regular_

### Official Review · Reviewer_M9fs · 2026-03-12

**Soundness:** 3
**Presentation:** 3
**Significance:** 3
**Originality:** 2
**Overall Recommendation:** 4
**Confidence:** 2

**Summary:**

This paper addresses identified and addresses challenges of decision transformer which are susceptible to average action trap in Real time Bidding(RTB) for ads . The paper separates candidate generation and selection while using Retrieval augmented and distribution action modeling to generate multiple bids instead of averaging and then uses value model for selection based on expected reward. The paper made a reasonable tradeoff on latency but still under RTB constraints to make it practical for real world deployment.

**Compliance With Llm Reviewing Policy:**

Affirmed.

**Ethical Review Flag:**

Flag this paper for an ethics review.

**Key Questions For Authors:**

Whats the size of the the RAG that was tested for ~29 GB memory and have you tested with index which does not fit in memory and how does that impact latency?

How is the GMM trained in the online setting (where bids are dynamic)?

What’s the minimum K that guarantees >90% recall of high-value bids ( high CTR/CPA)? How does this vary with ad quality?

**Limitations:**

Yes

**Strengths And Weaknesses:**

**Strengths**

Diagnosis of average action problem and proposing bid distribution

RAG for memory based RL is practical for production system

Decouping of candidate generation and selection is powerful and scalable

**Weakness**

Offline evaluations may not translate to real world outcomes, A/B tests would be beneficial to validate effectiveness in real world online applications

Real worls bidding is much more complex and are not modeled for ex. pacing, compaign constraints etc

---

> ### Author Rebuttal · Authors · 2026-03-31
>
> We thank the reviewer for the insightful comments. Our responses are as follows:
>
> **Q1:** Online Evaluation and A/B Testing.
>
> **A1:** Thanks for highlighting the importance of real-world A/B testing. Currently, we evaluated DRIVE on AuctionNet, a large-scale industrial benchmark constructed from real-world bidding logs that provides a representative environment under strict budget and CPA constraints. While we acknowledge the importance of real-world A/B testing, such implementation requires substantial system integration and infrastructure that are beyond the scope of the current research. We therefore follow the established convention of using high-fidelity offline benchmarks as the primary evaluation protocol. We hope that the strong offline performance demonstrated here will motivate future deployment studies in live auction systems.
>
> **Q2:**  Real-World Bidding Complexity.
>
> **A2:**  We agree that real-world bidding involves additional complexity beyond what any single benchmark can fully capture. DRIVE navigates pacing via Transformer-based sequence modeling with time_left and budget_left as core features to learn the temporal dependencies required for adaptive pacing across 48 decision steps. Additionally, campaign constraints such as target CPA are explicitly handled by our constraint-aware value critic via a shaped reward function (Eq. 13) that penalizes violations; as demonstrated in Appendix C.4 (Figure 9), this approach significantly reduces the CPA Exceed Rate compared to naive baselines, proving the framework's ability to better satisfy industrial requirements and its extensibility to various other production KPIs. Certain complexities in production systems, including non-stationary market distributions and richer constraint types, are not fully modeled by AuctionNet, and we view this as a promising direction for future work.
>
> **Q3:** What's the size of the RAG that was tested for ~29 GB memory and have you tested with index which does not fit in memory and how does that impact latency?
>
> **A3:** The 11 GB FAISS index consists of approximately 23 million state embeddings. with peak Resident Set Size of ~29 GB including model weights and data buffers, and an average retrieval latency of 11.86 ms. For memory-constrained environments, we validated an on-disk (mmap) mode that reduces RSS to 6.8 GB with negligible latency impact due to OS page caching. For extreme scaling, Product Quantization (PQ) can compress the index $8-16×$, keeping it memory-resident within 10-20 ms latency. This demonstrates DRIVE's deployment flexibility across different memory configurations.
>
> **Q4:** How is the GMM trained in the online setting (where bids are dynamic)?
>
> **A4:** Thanks for this question. We clarify that DRIVE is an **offline RL framework** trained exclusively on historical logs (Eq. 6) without online updates. Dynamic adaptability is achieved during inference because the Transformer backbone predicts GMM parameters ($\alpha_m, \mu_m, \sigma_m^2$) at each step based on real-time context, such as remaining budget and market state. This enables the policy to shift between bidding modes (e.g., aggressive vs. conservative) to adapt to fluctuations in real-time without requiring online retraining.
>
> **Q5:** What’s the minimum $K$ that guarantees >90% recall of high-value bids ( high CTR/CPA)? How does this vary with ad quality?
>
> **A5:** Thank you for this precise question. We do not claim that there exists a universal minimum $K$ that guarantees (>90%) recall of high-value bids across all ads or traffic conditions. The required retrieval depth depends on local action diversity and data density.
>
> In DRIVE, retrieval is not used in isolation. We first retrieve $3 \times K$ candidates by cosine similarity and then keep the top-$K$ according to stored RTG score, which biases the retained set toward higher-value historical actions. In addition, index pre-filtering excludes RTG=0 transitions at construction time (Appendix A.3). These design choices make the overall method less sensitive to the raw choice of $K$. Empirically, Figure 12 shows relatively stable performance over $K=1$ to $K=9$ (Average Value: 393–404), suggesting that even small $K$ can be effective in our benchmark once retrieval is combined with reranking.
>
> Regarding ad quality variation, the query embedding $h_t$​ explicitly incorporates pValue-related features, so FAISS retrieval naturally returns neighbors from historically similar ad quality contexts without requiring $K$ to vary explicitly.
>
> Thank you again for the insightful feedback. It has significantly improved the clarity and rigor of our work.

---

> > ### Author Rebuttal · Reviewer_M9fs · 2026-04-05
> >
> > My concerns have been addressed. Thank you for the response.

---

> > > ### Author Response · Authors · 2026-04-08
> > >
> > > Thank you very much for your thoughtful feedback! We sincerely appreciate your support and the time you invested in carefully reviewing and engaging with our work.

---

### Official Review · Reviewer_SdEU · 2026-03-12

**Soundness:** 3
**Presentation:** 3
**Significance:** 3
**Originality:** 2
**Overall Recommendation:** 5
**Confidence:** 3

**Summary:**

This paper introduces DRIVE, a transformer-based framework for offline reinforcement learning (principally in the setting of auto-bidding on advertisements). It is comprised of three main components: a GMM-based actor/policy network, a retrieval architecture (which uses an encoder transformer to generate embeddings of the train data), and an IQL-based critic network. The framework chooses actions as follows: at each step, it samples $L$ actions from its policy network, and $K$ actions from the retrieval architecture (that is, this step samples from actions in the dataset which performed well on similar states, where the similarity is based on what is learnt from the transformer encoder). It then ranks these $L+K$ actions using the value network, and proceeds with the highest-ranked action.

They evaluate this method on AuctionBench and the D4RL MuJoCo and Maze2D benchmarks, and compare against a range of baselines in three categories: classical offline RL, behaviour-cloning based methods, and auction-specific baselines. They further perform an ablation, evaluating how different parts of the framework affect performance when they are included/replaced/swapped out with a baseline.

**Compliance With Llm Reviewing Policy:**

Affirmed.

**Final Justification:**

I thank the authors' for the effort in their rebuttal and addressing my questions. As stated in my reply rebuttal comment, my questions were addressed, and I raised my score to a 5.

That being said, I also read the other reviews and discussions, and I chose to lower my confidence to a 3 -- this is because I am not an expert on the literature surrounding auto-bidding, so this reflects my uncertainty with regards to the novelty/potential impact of this work.

**Key Questions For Authors:**

- How much is runtime increased by doing the retrieval + reranking, compared to simply sampling from the policy? (Of course this will result in worse performance, but it might be worth having an idea of how much this increased performance "costs").
- Is the retrieval architecture necessary as the transformer is scaled up? I can imagine that at a certain point all training data is nearly memorized, so that the additional retrieval is not needed.

**Limitations:**

yes

**Strengths And Weaknesses:**

**Strengths**
- The argument for the need for multimodal action distributions is well-presented, both in a toy setting in Figure 1 and with real data in the appendix.
- The authors give a clear presentation of the setting (auto-bidding) and the sequence of prior work in this area, allowing readers from other domains to easily situate the paper in the context of previous work.
- I very much appreciate the extensive ablation done in Section 5 and Appendix C. Reading through it greatly built my intuition on to which degree different aspects improved performance, and I think that this will make the work more applicable to general practitioners.
- The authors consider a relatively wide range of environments on which they evaluate their setting, and compare to a good number of baselines.

**Weaknesses**
- Perhaps the largest weakness is the paper's relative lack of novelty: to my understanding, none of the components of DRIVE (that is GMM policy modelling, retrieval over latent representations, and ranking with an offline critic) are novel to this work. That isn't to say this work isn't novel in its own right (e.g. the Rainbow paper also combined existing improvements into a single algorithm).

---

> ### Author Rebuttal · Authors · 2026-03-31
>
> We sincerely thank the reviewer for the constructive and thoughtful feedback. Below, we address your concerns in detail:
>
> **Q1:** Limited novelty and the reason to use these techniques for the auto-bidding task. (weakness1 and weakness2)
>
> **A1:** We appreciate the reviewer’s insightful comparison to the Rainbow paper. The core novelty of DRIVE is not the invention of these components in isolation, but a **problem-driven integration** specifically engineered to resolve the **"Average Action Trap"** and **unreliability under long-tail traffic** in the auto-bidding task.
> We use the GMM head to explicitly capture the multimodal nature of optimal bids, which unimodal regression fails to model. RAG provides explicit non-parametric support to rectify model unreliability under sparse or long-tail traffic. The Value-Based Critic acts as a robust decision-maker to rank and select the final action.
> The synergy of these modules enables DRIVE to deliver robust and reliable bidding policies (As shown in Table 3).
>
> **Q2:** How much is runtime increased by doing the retrieval + reranking, compared to simply sampling from the policy?
>
> **A2:** While a standard Decision Transformer (DT) baseline takes **10.44 ms** per step, DRIVE records an average of **46.38 ms**, which is well within the 50 ms real-time bidding (RTB) constraint. We provide a detailed latency breakdown to quantify this "cost" :
>
> | **Component**         | **Description**                                            | **Latency (ms)** |
> | --------------------- | ---------------------------------------------------------- | ---------------- |
> | Transformer Inference | Two forward passes for GMM parameters & retrieval encoding | $27.15$          |
> | FAISS Retrieval       | Efficient nearest neighbor search in trajectory memory     | $11.86$          |
> | Value Critic          | Reranking candidates via the IQL-based Critic              | $7.37$           |
> | Total (DRIVE)         | Full end-to-end inference                                  | **$46.38$**      |
>
>
> **Q3:** Is the retrieval architecture necessary as the transformer is scaled up? I can imagine that at a certain point all training data is nearly memorized, so that the additional retrieval is not needed.
>
> **A3:** Thanks for the thought-provoking question regarding the potential of model scaling to replace retrieval. We contend that the performance gains of DRIVE are driven by a fundamental mechanistic shift rather than a simple increase in model capacity (Scaling Law). While scaling parameters can improve implicit memorization, it does not resolve the "Average Action" trap inherent to the unimodal regression objectives (e.g., MSE) used in most sequence modeling policies; even a massive Transformer will still gravitate toward suboptimal averaged actions when facing stochastic market modes.
>
> Furthermore, purely parametric models remain vulnerable to unreliable predicted actions in sparse or long-tail regions, where scaling alone cannot provide the explicit "historical evidence" that our retrieval module offers as a non-parametric anchor. Crucially, we validated this across structurally diverse backbones, including BC, CDT, and PDiT, which possess significantly different parameter counts and architectural designs. The consistent improvement across these varied scales and architectures (detailed in Table 8) demonstrates that the synergy of distributional modeling and retrieval helps mitigate inherent paradigm limitations that scaling parameters alone may not sufficiently address.
>
> We greatly appreciate your insightful comments, which have helped us improve both the clarity and completeness of our analysis.

---

> > ### Author Rebuttal · Reviewer_SdEU · 2026-04-04
> >
> > I thank the authors for their rebuttal, they have addressed all my questions. I am happy to increase my score to a 5.

---

> > > ### Author Response · Authors · 2026-04-08
> > >
> > > Thank you very much for taking the time to revisit our work during the rebuttal phase and for increasing your score! We greatly appreciate your suggestion, which helped us better articulate both the technical motivation and the connections between our proposed components.

---

### Official Review · Reviewer_rsvW · 2026-03-13

**Soundness:** 2
**Presentation:** 3
**Significance:** 2
**Originality:** 2
**Overall Recommendation:** 3
**Confidence:** 4

**Summary:**

This paper studies offline auto-bidding and argues that standard DT-style methods are limited by unimodal action prediction and purely parametric generation, which can lead to averaged, suboptimal bids and poor behavior in sparse or long-tail regions. The proposed method, DRIVE, combines three components: a GMM-based action head to model multimodal actions, a retrieval module that fetches high-quality historical actions from similar contexts, and an IQL-style critic that ranks generated and retrieved candidates at inference time. The method is evaluated on AuctionNet, D4RL Gym, and Maze2D tasks, and the paper reports improved performance over several offline RL and bidding-specific baselines. The paper also includes ablations and a plug-and-play study across several Transformer backbones.

**Compliance With Llm Reviewing Policy:**

Affirmed.

**Final Justification:**

Thank you for the author's rebuttal, additional explanations, and experiments. Overall, I think the paper needs a major revision and falls short of the bar for acceptance. Specifically,

(1)  GMM is a common method to increase the multimodality of the distribution.

(2) The proposed RAG method seems to have a theory-to-practice gap. Specifically, does "similar state embedding" mean "similar optimal action"? Perhaps, there should be a guarantee such that $||a_1^{opt}-a_2^{opt}||\le \alpha || stateEmb_1-stateEmb_2||$.

(3) About the multimodality of the Q function. I think one should measure the multimodality of the optimal Q function, $Q^*$, to assess its multimodal nature.

(4) Besides, unlike Rainbow, which investigates many useful components in RL and tightly couples them into a method that fundamentally advances RL, DRIVE does not study many of the existing methods or tricks in auto-bidding, but adopts three other methods to construct a method. I think the analogy between Rainbow and DRIVE is not very appropriate.

**Key Questions For Authors:**

1. Can you provide analysis or experimental evidence that the diffusion model action predictor is not as good as the Transformer with GMM?

2. Can you provide a quantitative measure of multimodality or mean-action failure beyond the examples in Figure 2? For example, how often does the deterministic DT prediction fall in a low-density region of the local action distribution, and how strongly does that correlate with return degradation? A strong answer here would materially strengthen the paper.

**Limitations:**

yes

**Strengths And Weaknesses:**

**Strengths:**
1. The paper studies an important industrial problem, the auto-bidding problem, which is of vital real-world significance.
2. The paper discovers an important "average action" trap that can be widely observed in practice.
3. Extensive simulated experiments on both auto-bidding and general tasks are conducted, which validate the effectiveness of the proposed method.

**Weaknesses:**
1. The motivation for designing GMM should be further clarified. Generally, diffusion models are known for their multimodal learning capabilities, which can help avoid the "average action" trap. The paper adopts an end-to-end action prediction paradigm, and one possible method is to use a diffusion model to directly predict actions (rather than diffBid, which adopts a different planning-and-control architecture), a setting that has been studied in previous works. The motivation seems to omit this.

2. The approach design is not principled. The proposed method combines three specific methods that seem practical but are heuristic. However, some theoretical analysis for motivation derivation, method design, or convergence analysis would be appreciated.

3. The paper reports no online experiment results. Real-world experiments would also be appreciated. (However, this is not the main concern.)

---

> ### Author Rebuttal · Authors · 2026-03-31
>
> We sincerely thank the reviewer for the thoughtful and constructive comments! Below, we respond to your concerns point by point:
>
> **Q1:** Motivation for GMM head and experimental evidence with diffusion model action predictor. (Weakness 1 and Key Question 1)
>
> **A1:** Thank you for raising this important point! We chose a GMM-based head rather than Diffusion to balance **modeling parsimony, performance, and industrial feasibility.** The auto-bidding action ($\lambda_t$) is 1-dimensional, continuous but typically exhibits a multimodal structure. GMM captures this via a closed-form mixture likelihood, avoiding the "Average Action Trap" without Diffusion’s generative overhead. Empirically, we conduct a controlled experiment replacing the GMM action head with a Diffusion-based action head (DDPM, T=100 steps), while keeping the Transformer backbone and IQL Critic identical. As shown in the table below, GMM consistently matches or exceeds the Diffusion head's performance:
>
> | Budget(%) | GMM Gen. + Critic (No Retr.) | Diffusion head+Critic |
> | ------------- | ------------------------------- | -------------------------- |
> | 50               |      **205.4 ± 3.14**             | 190.7 ± 2.21          |
> | 75               |      **296.7 ± 2.39**             | 273.2 ± 1.82          |
> | 100             |      **378.4 ± 2.68**             | 376.3 ± 5.72          |
> | 125             |        461.3 ± 4.62                 | **467.2 ± 2.53**      |
> | 150             |     **532.2 ± 2.16**             | 531.4 ± 4.05          |
> | Average      |              374.8                        | 367.8                 |
>
> Crucially, the choice of a GMM head is necessitated by the strict latency requirements of Real-Time Bidding (RTB). We report the sampling latency comparison below:
>
> | Method  | GMM Head（No Retrieval） | Diffusion Head (DDPM, T=100) |
> | -----| --------------------- | ---------------------------- |
> | Latency | 11.03 ± 0.39 ms       | 223.32 ± 5.55ms              |
>
> These results confirm that GMM achieves a superior **performance and efficiency trade-off**, making it the principled choice for this industrial setting rather than a compromise.
>
> **Q2:** The design of DRIVE appears heuristic and lacks a principled foundation or theoretical motivation. (Weakness 2)
>
> **A2:** We clarify that DRIVE is not a heuristic combination of modules, but a problem-driven integration specifically engineered to mitigate the key challenges in offline auto-bidding:
> (1) GMM Head: Effectively handles the "Average Action Trap" by extending the standard unimodal assumption to support multi-modal strategies commonly found in auction data.
> (2) Retrieval: Mitigates parametric unreliability in long-tail state regions. By decoupling historical evidence from the Transformer's generalization error, it stabilizes decisions where data support is low.
> (3) Value Critic: Provides a unified evaluation signal via IQL. This allows for optimal selection from the hybrid candidate pool while mitigating the risk of OOD actions.
> As shown in the ablation study (Table 3), these modules yield complementary gains by mitigating distinct failure modes, providing an alternative to standard sequence modeling in offline RL.
>
> **Q3:** Online Experiment Results. (Weakness 3)
>
> **A3:** Thank you for highlighting the importance of online deployment. While we acknowledge that real-world A/B testing is the ultimate validation, such implementation requires substantial system integration and infrastructure beyond the current research scope. We therefore utilize AuctionNet, a large-scale benchmark derived from real-world bidding logs as a high-fidelity proxy under strict budget and CPA constraints, and view online deployment as a valuable future direction.
>
> **Q4:** A quantitative measure of how frequently deterministic DT predictions fall into low-density regions and how this correlates with return degradation. (Key Question 2)
>
> **A4:** In the anonymous link (https://anonymous.4open.science/r/DRIVE_11512_analyze_picture/Appendix.md), we provide two figures to address this query. As shown in Figure 13, we define the local action distribution using $K=400$ nearest-neighbor states retrieved from the offline dataset via our RAG module. We define Low Density Regions (LDR) as a percentile less than 10% of the action density estimated via KDE. Across all test periods, the DT suffers from a high LDR rate of 58.2% to 69.2%.  Figure 14 correlates this value-guided selection with significant gains, yielding a Mean Return Gap of +96.2 and an 87.8% Win Rate over 336 episodes. This demonstrates that steering the policy away from low-density "average actions" toward high-value regions is the foundational driver of DRIVE's superior performance in industrial bidding scenarios.
>
> We deeply appreciate your constructive feedback. Your insights significantly enhance the completeness of our work, and we hope these improvements will help raise your evaluation.

---

> > ### Author Rebuttal · Reviewer_rsvW · 2026-04-04
> >
> > Thank you for the rebuttal and the additional experiments. However, I still have major concerns.
> >
> > (1) From response 1, it seems that the GMM-based head, the main proposed method in the paper, appears to be motivated only by engineering feasibility rather than by clear theoretical advantages. Given that GMM is a basic method, I think this design makes a limited academic contribution.
> >
> > (2) The complete algorithm is a combination of three existing methods. Although the paper can be a good technical report in auto-bidding, the academic novelty is limited.
> >
> > (3) I am still confused about how severe the"mean action" trap, one of the main motivations of the paper,  is. For example, can we statistically measure the degree of multimodality of the Q function?
> >
> > Overall, after reading other reviews and the author's responses, I still think this paper falls short of the bar for acceptance.

---

> > > ### Author Response · Authors · 2026-04-05
> > >
> > > We sincerely appreciate the reviewer's continued engagement and address your concerns directly.
> > >
> > > Q1: On the theoretical motivation for GMM.
> > >
> > > A1: We respectfully argue that the GMM head is problem-driven but also has some statistical learning theory to support.
> > > MDN [1] establishes that under standard squared-error regression, the optimal point predictor **provably converges to the conditional mean  $\mathbb{E}[a|o]$**. IBC [2] further formalizes this by showing that explicit single-output models must traverse intermediate values under multimodal supervision, _while demonstrating that  MDN-style parameterizations can already be highly effective in low-dimensional action spaces and attributing the motivation for implicit EBMs primarily to optimization challenges in **higher-dimensional settings**_. Since auto-bidding involves a **1D action space**, GMM provides a **sufficient and principled** distributional representation, consistent with the empirical findings of IBC in 1D settings.
> > >
> > > [1] Bishop et al. Mixture density networks.
> > >
> > > [2] Florence et al. Implicit behavioral cloning. CoRL 2022.
> > >
> > > Q2: A combination of three existing methods.
> > >
> > > A2: While Rainbow [3], a highly influential work in deep reinforcement learning, demonstrates that the principled integration of complementary components, each addressing a distinct limitation of the base algorithm, constitutes a meaningful academic contribution in its own right, DRIVE follows and extends this paradigm in an important way. Rather than combining independent and orthogonal modules, DRIVE integrates three interdependent components into a cohesive framework to address the "Average Action Trap" and long-tail unreliability in DT-style offline bidding, with the Value Critic acting as a principled fusion mechanism to select the optimal action from the hybrid candidate pool. The academic contribution therefore lies in identifying two structural limitations inherent to DT-style sequence modeling that existing methods cannot simultaneously resolve, and in designing a framework where each component plays a necessary and interdependent role, as directly validated by the ablation study in Table 3 and confirmed across diverse Transformer backbones in Table 8.
> > >
> > > [3] Hessel et al. Rainbow: Combining Improvements in Deep Reinforcement Learning. AAAI 2018.
> > >
> > > Q3: The multimodality of the Q function.
> > >
> > > A3: To directly address the reviewer's question on the degree of multimodality of the Q-function, we conduct a new experiment on 2,000 randomly sampled states from the test set, evaluating $Q(s,a)$ over a uniform grid of 100 action points spanning the full action range. We classify each state by the number of local peaks in its Q-function profile and report DT action quality for each group, as shown in the table below. Additionally, we provide per-period results in the anonymous link( https://anonymous.4open.science/r/DRIVE_11512_analyze_picture/Appendix.md). Here, the suboptimal rate is defined as the proportion of decision steps where the DT action's Q-percentile rank falls below 80% among all 100 uniformly sampled actions. DT Distance to $a^∗$ measures the absolute difference between the DT output action and the optimal action $a^∗ =\arg⁡ max⁡ Q(s,a)$ in the current state.
> > >
> > > | Q Shape                   | Proportion | Suboptimal Rate | DT Distance to $a^∗$ |
> > > | ------------------------- | ---------- | --------------- | -------------------- |
> > > | Unimodal (1 peak)         | 82.4%      | 38.2%           | 4.24                 |
> > > | **Multimodal (≥2 peaks)** | **17.6%**  | **54.7%**       | **6.10**             |
> > > | Overall                   | 100%       | 41.1%           | 4.51                 |
> > >
> > >  The results reveal a clear and statistically significant pattern. In the dominant unimodal regime (82.4%), DT's mean-regression naturally aligns close to the single optimal peak, yielding a moderate 38.2% suboptimal rate. However, in the multimodal regime (17.6% of states), where the Q-function exhibits multiple local peaks, **the suboptimal rate rises sharply to 54.7%**, nearly $1.5×$ higher than the unimodal case, and the average distance to $a^∗$ increases from 4.24 to 6.10, confirming that DT's mean-regression output falls precisely into the suboptimal valley between Q-function modes. These results directly quantify the severity of the Average Action Trap, while it remains moderate in the dominant unimodal regime, it becomes a severe and systematic failure in multimodal states, which constitute a non-trivial 17.6% of the decision space. We note that the multimodality of the action distribution has been addressed in earlier rebuttal response, supported by empirical evidence in Figure 13 of the anonymous link.
> > >
> > > We hope these additional theoretical justifications and quantitative analyses have adequately addressed the concerns raised, and we remain committed to further improving the clarity and rigor of our work.

---

### Official Review · Reviewer_Stzw · 2026-03-29

**Soundness:** 3
**Presentation:** 3
**Significance:** 2
**Originality:** 2
**Overall Recommendation:** 2
**Confidence:** 4

**Summary:**

The paper proposes DRIVE, an enhancement on top of decision transformers designed for the task of auto-bidding. The method is a combination of three components: distributional action modeling (use mixture of Gaussians instead of deterministic actions), retrieval-augmented candidate generation, and value-based action selection. Experiments on AuctionNet and additional offline reinforcement learning benchmarks demonstrate good performance.

**Compliance With Llm Reviewing Policy:**

Affirmed.

**Key Questions For Authors:**

1. Regarding Figure 1, why do this multi-modal phenomenon happen in the auto-bidding task? Taking budget constrained bidding as an example, for a state s and two actions (bidding parameters) a_1 < a_2, consider the average action a_3=(a_1+a_2)/2, the immediate reward of a_3 should be in between a_1 and a_2. The budget consumption of a_3 should also be in between. Therefore, the Q value of a_3 should also be in between. In this example, there is no multi-modality.

**Limitations:**

Yes

**Strengths And Weaknesses:**

Strengths
- The method is sound. All three components in DRIVE are appropriate improvements upon decision transformers, and are aligned with intuition.
- The presentation is clear.
- Experiments are conducted on both AuctionNet and D4RL datasets. The proposed method shows performance gain over vanilla DT, and achieves best performance in some settings.

Weakness
- The main weakness is the limited novelty of the proposed method. All three parts of DRIVE are not new, and the paper seems to combine the three techniques without novel insights or analysis:
 1. Using GMMs as an expressive policy class for RL is not new (e.g., [1,2]). In fact, one can plug in any probabilistic model (e.g., diffusion models) as an RL actor.
 2. As stated by the authors, there are recent work that incorporate retrieval-augmented generation into DT.
 3. Value-based action evaluation and selection for DT is an existing technique [3].

- Though the focus of the paper is the specific task of auto-bidding, the proposed improvement over DT seems to be applicable to general offline RL tasks, e.g., there are no domain-specific considerations in the method. It will be better to have more discussions on the reason to use these techniques for the auto-bidding task.

[1] Baram et al. Maximum Entropy Reinforcement Learning with Mixture Policies.
[2] Dey et al. Comparing Deterministic and Soft Policy Gradients for Optimizing Gaussian Mixture Actors. TMLR 2024.
[3] Hu et al. Q-value Regularized Transformer for Offline Reinforcement Learning. ICML 2024.

---

> ### Author Rebuttal · Authors · 2026-03-31
>
> We sincerely thank the reviewer for the thoughtful and detailed comments. Below, we respond to the key points raised in the "Weaknesses" and "Key Questions" sections:
>
> **Q1:** Limited novelty and the reason to use these techniques for the auto-bidding task. (Weakness1 and Weakness2)
>
> **A1:** Thanks for the assessment. We clarify that DRIVE’s primary contribution is not the invention of individual modules, but a **problem-driven integration** specifically engineered to resolve the **"Average Action Trap"** and **unreliability under long-tail traffic** in the auto-bidding task.
>
> - Thanks for highlighting the relevant works [1, 2]. These two works leverage GMMs for **online** exploration to escape local optima, while DRIVE operates in an **offline** setting where the GMM head represents $M$ distinct bidding strategy distributions from logged data to avoid suboptimal "average" actions. Diffusion-based models remain impractical due to their excessive sampling latency, which is prohibitive in Real-Time Bidding (RTB). Our sampling experiments confirm this limitation, with the GMM head requiring only **11.03 ms** (sampling + critic, without retrieval), whereas the Diffusion-based head (DDPM, $T=100$) under the same setting incurs a **223.32 ms** latency.
> - Compared with previous methods that use retrieval for context enhancement, RAG in our work enables the policy to anchor its decisions to previously observed high-performing behaviors, particularly in sparse and longtail bidding scenarios. This design improves the robustness of decisions through historical real evidence.
> - Regarding QT [3], we acknowledge the importance of value assessment. In DRIVE, the Value Critic is further designed as a selector to optimally screen the mixed candidate pool composed of GMM and RAG. Employing a constraint-aware reward function (Eq. 13) provides a specialized mechanism to better align decisions with industrial constraints like  CPA, thereby enhancing decision reliability in complex bidding environments.
>
> [1] Baram et al. Maximum Entropy Reinforcement Learning with Mixture Policies.
>
> [2] Dey et al. Comparing Deterministic and Soft Policy Gradients for Optimizing Gaussian Mixture Actors. TMLR 2024.
>
> [3] Hu et al. Q-value Regularized Transformer for Offline Reinforcement Learning. ICML 2024.
>
> **Q2:** Why does this multi-modal phenomenon happen in the auto-bidding task?
>
> **A2:** Thank you for this insightful question regarding the unimodal assumption. We agree that the reviewer’s reasoning regarding unimodality is valid for **static and single-step** scenarios, where a higher action means a higher value. However, in sequential budget-constrained bidding, the relationship between action and $Q$-values is non-monotonic because **$Q$-values represent expected long-term cumulative conversions** (e.g., clicks or orders), not just immediate wins. While a higher bid $a_2$ increases immediate winning probability, it also accelerates budget depletion, potentially sacrificing future high-value opportunities that a conservative bid $a_1$ could have captured. Thus, $Q(a_1)$ can be higher than or equal to $Q(a_2)$, depending on the remaining budget and time.
>
> Since both $a_1$​ and $a_2$ can be optimal actions under the same state, depending on the advertiser's strategies and risk preferences, they co-exist in the offline dataset under identical states, naturally producing a **multimodal action distribution**. A unimodal policy regresses toward their weighted mean $a_3$​, which falls into the "suboptimal valley" — too low to secure impressions targeted by $a_2$, yet too expensive to maintain the cost efficiency of $a_1$​, consequently collapsing into the "Average Action Trap." In our paper, we show some empirical failure cases in Figure 2 and Figure 7.
>
> Thank you again for your valuable questions, which have helped us clarify the foundational drivers of our framework. We hope these technical clarifications and empirical evidence address your concerns and help raise your evaluation of our work.

---

### Decision · Program_Chairs · 2026-04-30

**Decision:**

Accept (regular)

**Comment:**

The paper introduces an offline reinforcement learning framework for auto-bidding that mitigates the average action trap by integrating a GMM-based action head, retrieval-augmented generation, and a value critic. I find the proposed integration of 3 components very practical, and reviewers also mentioned that the proposed method is sound and well presented. Although there were a few concerns about novelty of the proposed components, I feel the work will be valuable to the research community. Other technical and evaluation concerns raised by reviewers are addressed to a great extent in the rebuttal. Overall, I recommend weak accept.